# Reticulon proteins modulate autophagy of the endoplasmic reticulum in maize endosperm

Xiaoguo Zhang[1†], Xinxin Ding[1†], Richard Scott Marshall[2†], Julio Paez-Valencia[1], Patrick Lacey[1], Richard David Vierstra[2], Marisa S Otegui[1,3*]

[1]Department of Botany, Laboratory of Cell and Molecular Biology, University of Wisconsin, Madison, United States; [2]Department of Biology, Washington University in St. Louis, St. Louis, United States; [3]Department of Genetics, University of Wisconsin, Madison, United States

**Abstract** Reticulon (Rtn) proteins shape tubular domains of the endoplasmic reticulum (ER), and in some cases are autophagy receptors for selective ER turnover. We have found that maize Rtn1 and Rtn2 control ER homeostasis and autophagic flux in endosperm aleurone cells, where the ER accumulates lipid droplets and synthesizes storage protein accretions metabolized during germination. Maize Rtn1 and Rtn2 are expressed in the endosperm, localize to the ER, and re-model ER architecture in a dose-dependent manner. Rtn1 and Rtn2 interact with Atg8a using four Atg8-interacting motifs (AIMs) located at the C-terminus, cytoplasmic loop, and within the transmembrane segments. Binding between Rtn2 and Atg8 is elevated upon ER stress. Maize *rtn2* mutants display increased autophagy and up-regulation of an ER stress-responsive chaperone. We propose that maize Rtn1 and Rtn2 act as receptors for autophagy-mediated ER turnover, and thus are critical for ER homeostasis and suppression of ER stress.

**\*For correspondence:**
otegui@wisc.edu

[†]These authors contributed equally to this work

**Competing interests:** The authors declare that no competing interests exist.

## Introduction

The endoplasmic reticulum (ER) is the most extensive and versatile organelle in plants, consisting of a complex three-dimensional network of tubules and cisternae within each cell, and projections that provide cell-to-cell connections through plasmodesmata. Besides being a site of synthesis, folding, and modification of proteins, lipids, and hormones, and the entry point for nascent proteins into the secretory system (*Stefano and Brandizzi, 2018*; *Helliwell, 2006*), the ER participates in calcium-mediated signaling, and the perception and response to protein folding stress (*Howell, 2013*; *Srivastava et al., 2018*).

ER stress occurs in response to a number of physiological and pathological conditions that saturate ER protein folding capacity, leading to the accumulation of misfolded proteins. As a consequence of ER stress, eukaryotic cells: i) up-regulate expression of stress-responsive genes to restore protein-folding capacity in the ER *via* a mechanism called the unfolded protein response (UPR) (*Hetz, 2012*); ii) expand ER volume to accommodate the increased protein load (*Schuck et al., 2009*); iii) promote ER-associated degradation (ERAD) of unfolded or misfolded proteins through the ubiquitin-proteasome system (*Mehrtash and Hochstrasser, 2019*); and iv) trigger turnover of selected ER domains via autophagy (*Rashid et al., 2015*).

The ER is constantly adjusting its size, shape, and activity in response to developmental cues and cellular demands. Its dynamic shape is controlled by a large number of proteins (*Hu et al., 2008*; *Hu et al., 2011*), including the reticulon (RTN) protein family (*Voeltz et al., 2006*; *Nziengui and Schoefs, 2009*). RTNs contain a signature reticulon homology domain (RHD) with two major hydrophobic segments forming a pair of V-shaped transmembrane wedges joined by a cytosolic loop,

with both the N- and C-termini facing the cytosol (*Breeze et al., 2016*; *Kriechbaumer et al., 2015*). RTNs are generally required for the formation of ER tubules, although some RTNs preferentially locate to ER cisternal edges (*Khaminets et al., 2015*). The mechanisms that control ER homeostasis and restore ER normal size upon cessation of ER stress are not completely understood (*Loi et al., 2018*), but appear to involve autophagy and RTNs in both animals and yeast (*Bernales et al., 2006*).

Selected portions of the ER and other organelles can be degraded through autophagy. In plants, two major autophagic routes have been identified: macro- and micro-autophagy (*Ding et al., 2018*). During macro-autophagy, a cup-shaped, double membrane structure called the phagophore emerges from the ER (*Zhuang et al., 2017*), expands, and sequesters cytoplasmic contents as it closes to form a sealed autophagosome. The autophagosome then fuses with the tonoplast to release the internal membrane-bound core as an autophagic body into the vacuolar lumen where it is catabolized by vacuolar hydrolases (*Feng et al., 2014*). Macro-autophagy is mediated by multiple AUTOPHAGY-RELATED (ATG) proteins. Among them, members of the ATG8 family (known as MAP1LC3 or GABARAP in mammals) are critical components for autophagosome assembly and cargo selection. Upon autophagy induction, ATG8 becomes conjugated to phosphatidylethanol-amine and is subsequently incorporated into the inner and outer phagophore membranes, where it participates in phagophore expansion and maturation (*Weidberg et al., 2011*; *Yu and Melia, 2017*), tethering of appropriate autophagic cargo through its association with cargo receptors (*Zaffagnini and Martens, 2016*), and fusion of autophagosomes with lysosomes or vacuoles (*Nguyen et al., 2016*). Like in other members of the ubiquitin-fold superfamily, ATG8 conjugation involves the E1 activating enzyme ATG7, the E2 conjugating enzyme ATG3, and an E3 ligase com-plex, which consists of a conjugate of ATG12 and ATG5 bound to their partner ATG16 (*Kaufmann et al., 2014*; *Walczak and Martens, 2013*; *Noda et al., 2013*; *Romanov et al., 2012*; *Hanada et al., 2007*; *Chung et al., 2010*). ATG8 engages autophagy receptors either through ATG8-interacting motifs (AIMs; known as LC3-interacting regions or LIRs in mammals *Noda et al., 2010*) or through recently discovered ubiquitin-interacting motif (UIM)-like sequences (*Marshall et al., 2015*; *Marshall et al., 2019*), thus tethering the receptors and their associated cargo to autophagic membranes. During micro-autophagy, the tonoplast directly engulfs cargo by internalizing and pinching off vacuolar membrane domains, thus directly trapping cytoplasmic cargo in autophagic bodies, which are either stably stored or degraded inside the vacuole (*Müller et al., 2000*; *Sakai et al., 1998*; *Chanoca et al., 2015*). Both macro- and micro-autophagy can be either non-selective (bulk autophagy) or highly selective for specific cargo depending on physiological con-ditions (*Marshall and Vierstra, 2018*; *Ding et al., 2018*).

The ER is connected to autophagy in at least three distinct capacities; i) the ER provides mem-brane for building phagophores (*Lamb et al., 2013*; *Zhuang et al., 2017*); ii) autophagy is triggered upon ER stress to prevent cell death (*Ogata et al., 2006*); and iii) the ER is itself a target for selective autophagy. This selective autophagic removal of the ER, called ER-phagy (or reticulophagy), is trig-gered by ER stress and other stimuli such as starvation. In yeast, starvation induces Atg8-dependent macro-ER-phagy, whereas perturbations of ER redox homeostasis and the UPR lead to engulfment of ER sub-domains directly by the vacuole *via* a micro-ER-phagy route independent of Atg8 (*Schuck et al., 2014*; *Schäfer et al., 2020*; *Loi et al., 2019*).

In the past few years, multiple macro-ER-phagy receptors, some of which harbour RHD sequen-ces, have been identified in both yeast and mammals whereas only one receptor was identified in plants. In yeast, the ER-resident proteins Atg39 and the RHD-containing protein Atg40 (*Figure 1A*) directly interact with Atg8 to mediate ER-phagy of the nuclear envelope and cortical ER membranes, respectively, under starvation conditions (*Mochida et al., 2015*). Atg40 in particular interacts with Lunapark protein 1 (Lnp1), which resides at three-way tubular junctions of the ER, to bring Atg40-containing ER regions to the sites of phagophore initiation (*Chen et al., 2018*). Atg40 also interacts with components of the COPII assembly machinery to sort ER domains into autophagosomes (*Cui et al., 2019*). In mammals, the pair of RHD-containing proteins RTN3L (the long isoform of RTN3) (*Grumati et al., 2017*) and FAM134B (*Khaminets et al., 2015*; *Bhaskara et al., 2019*) act as ER-phagy receptors (*Figure 1A*), together with four other ER-resident proteins that lack RHD domains, namely CCPG1 (*Smith et al., 2018*), SEC62 (*Fumagalli et al., 2016*; *Chino et al., 2019*), ATL3 (*Chen et al., 2019*), and TEX264 (*Chino et al., 2019*; *An et al., 2019*). All reside in the ER, bind ATG8/MAP1LC3/GABARAP via AIMs or GIMs (GABARAP interacting motifs), and are required for cargo selection during ER-phagy. In plants, *Arabidopsis thaliana* SEC62 seems to play a

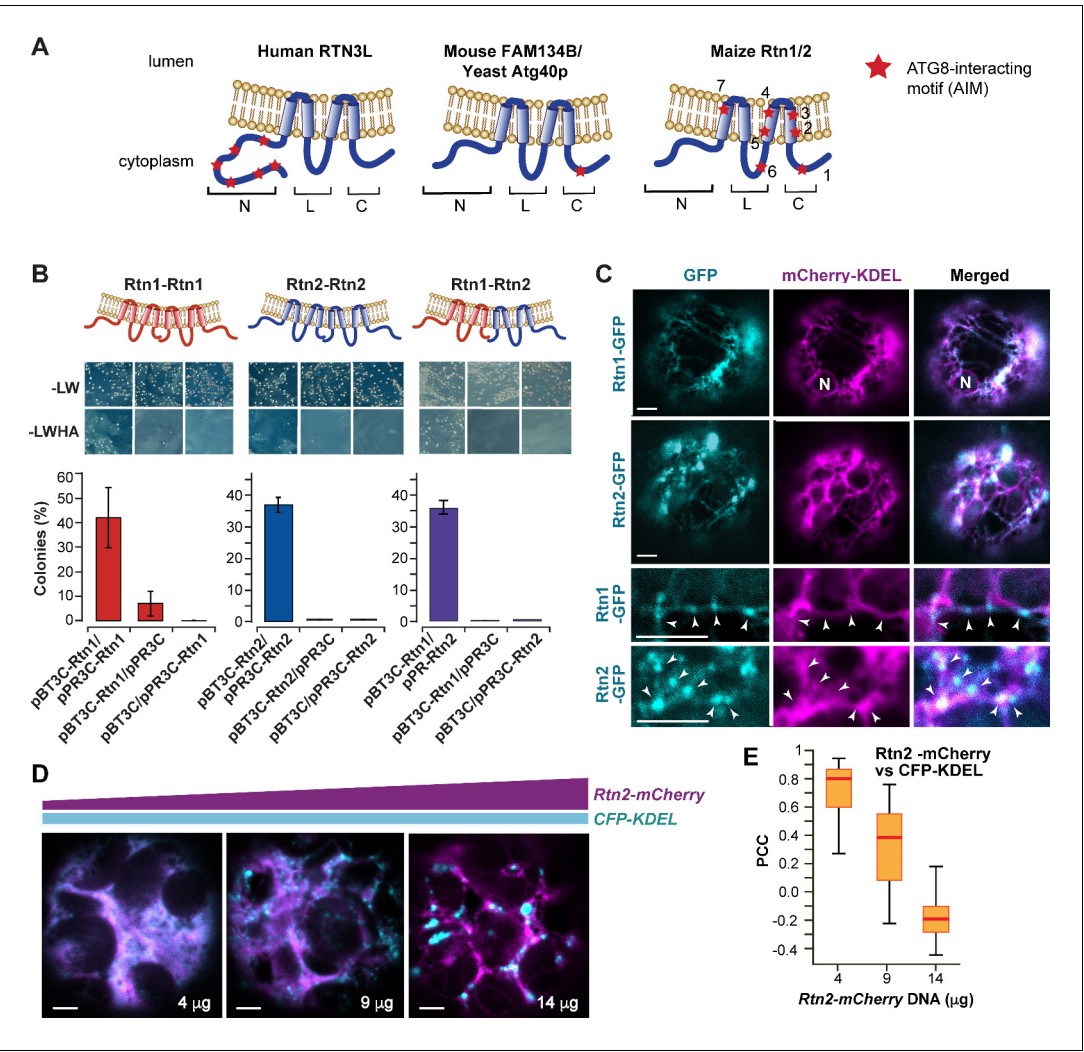

**Figure 1.** Maize Rtn1 and Rtn2 are reticulon proteins. (**A**) Diagrams of reticulon proteins with functions in ER-phagy. The position of functional AIMs in RTN3L and FAM134B/Atg40p and of predicted AIMs in maize Rtn1/2 are marked by red stars. (**B**) Split-ubiquitin Y2H assay showing homo- and hetero- interactions between Rtn1 and Rtn2. Plates show colonies grown on −LW medium (selecting for transformation) and on −LWHA medium (selecting for interaction). Negative controls were performed by co-expressing Rtn1 or Rtn2 with the corresponding empty vector. Bar graphs show ratios between the number of colonies grown on −LWHA medium versus −LW medium. The ratios from control plates were subtracted from those testing direct interactions between the Rtn proteins. Between 200 and 300 colonies were counted in each case. The graphs show the average of three independent experiments; error bars indicate SD. (**C**) Rtn1 and Rtn2 fused to GFP colocalize with the ER marker mCherry-KDEL in maize mesophyll protoplasts. The two lower panels show details of the preferential localization of Rtn1-GFP and Rtn2-GFP to tubular ER junctions (arrowheads). (**D**) Increasing expression of Rtn2-mCherry leads to the architectural reorganization of the ER. *Arabidopsis* protoplasts were transformed with increasing amounts (4, 9, and 14 µg) of plasmid containing *35S:Rtn2-mCherry* along with a constant amount of an independent plasmid encoding the ER lumenal marker CFP-KDEL. (**E**) Overlapping localization of Rtn2-mCherry and CFP-KDEL as shown by box and whisker plots of Pearson correlation coefficients (PCC) calculated for the protoplasts as shown in (**D**). The experiment was performed three times; graph shows the results of a representative experiment. Between 8 and 10 randomly selected protoplasts for each treatment were included in the analysis. Scale bars = 5 µm.
The online version of this article includes the following figure supplement(s) for figure 1:

**Figure supplement 1.** Structure and organization of plant reticulons.
**Figure supplement 2.** Expression patterns of reticulons in maize and rice.

conserved role as an ER-phagy receptor (*Hu et al., 2020*). However, the identities of other plant ER-phagy receptors, as well as their regulation and molecular mechanisms of action, are yet not known.

The ER in seed tissues such as the endosperm, is frequently specialized in the synthesis and accumulation of storage compounds metabolized during seed germination. In the cereal endosperm, two cell types with different storage functions arise during seed development: the starchy endosperm, which constitutes the bulk of the endosperm, and the peripheral aleurone cells (*Olsen, 2004*; *Becraft and Gutierrez-Marcos, 2012*; *Sabelli and Larkins, 2009*). The ER is extensively modified in both cell types; the starchy endosperm accumulates protein accretions called protein bodies inside the ER lumen, whereas the aleurone accumulates lipid droplets between the phospholipid leaflets of the ER membrane, both of which are consumed during seed germination (*Larkins and Hurkman, 1978*; *Pedrazzini et al., 2016*; *Bowman et al., 1988*; *Reyes et al., 2011*). Maize aleurone cells also transiently accumulate protein accretions in the ER, but deliver them to vacuoles by a poorly understood Atg8-independent autophagic mechanism (*Reyes et al., 2011*).

To help understand the functions of reticulon proteins in ER dynamics of cereal endosperm cells, we performed an in silico expression analysis and found that two highly similar RHD proteins, Rtn1 and Rtn2, are expressed in the maize endosperm cells during the accumulation of storage proteins and lipids. Here, we analyzed the functions of Rtn1 and Rtn2 using genetic, biochemical, and imaging analyses, and found that both act as AIM-containing ER-phagy receptors that protect maize aleurone cells against ER stress. The binding of Rtn proteins to Atg8 is positively regulated by ER stress, thus providing a mechanism to help promote ER turnover.

## Results

### Identification of Rtn proteins in maize

To identify Rtn proteins with potential functions in ER dynamics and autophagic turnover during the accumulation of storage proteins and lipids in maize endosperm cells, we first examined the phylogenetic relationships of the Rtn family proteins in maize and other plants. Whereas yeast and mammals have only two and four loci encoding RHD-containing proteins, respectively (*Oertle et al., 2003*), plants harbour large gene families with predicted RHD sequences (*Nziengui et al., 2007*; *Kriechbaumer et al., 2018*; *Nziengui and Schoefs, 2009*). For example, by using *Arabidopsis* RHD sequences in BLAST searches, we detected 23 and 18 genes encoding putative Rtn proteins in the maize (*Zea mays*) B73 and rice (*Oryza sativa*) genomes, respectively. Similar to the situation in mammals (*Di Scala et al., 2005*; *Oertle et al., 2003*), most maize *Rtn* genes generate multiple transcripts (AGPv4, Gramene). Using the longest transcript for each locus, we identified a large Rtn clade containing Rtn1-18 (clade 1) (*Figure 1—figure supplement 1A and B*). Proteins in this clade range from 133 to 283 amino acids and contain an RHD of approximately 155 amino acids flanked by short (less than 100 amino acids) N- and C-terminal extensions (*Nziengui et al., 2007*). A second clade (clade 2) contains both small and long Rtn proteins, including the long *Arabidopsis* RTNLB19 and 20 that harbor an N-terminal domain predicted to be involved in lipid synthesis (*Kriechbaumer et al., 2018*), and the small *Arabidopsis* RTNLB12 (203 amino acids) and maize Rtn19, 20, and 21 (186, 210, and 216 amino acids, respectively; *Figure 1—figure supplement 1B*). A third clade (clade 3) contains mostly long Rtn sequences such as maize Rtn22 (652 amino acids) and Rtn23 (439 amino acids).

Within clade 1, we identified a subgroup (clade 1–1) of maize and rice Rtn proteins that appeared restricted to cereals based on the absence of clear *Arabidopsis* paralogs (*Figure 1—figure supplement 1B*), and were among the most highly expressed in developing kernels (*Figure 1—figure supplement 2A,B*). Clade 1–1 included maize Rtn1 (Zm00001d043551) and Rtn2 (Zm00001d012776), which were expressed in the starchy endosperm and aleurone at 18–22 days after pollination (*Figure 1—figure supplement 2A*), coinciding with the time when storage compounds accumulate in the ER. *Rtn1* generates three different transcripts (AGPv4, Gramene) that code for either a 257-amino-acid protein with a predicted 72-residue N-terminal extension (*Figure 1—figure supplement 2C*); a 157-residue protein with no N-terminal extension and only half of the first transmembrane segment; or a 101-residue polypeptide with no predicted transmembrane domains. *Rtn2* generates only one transcript (AGPv4, Gramene) coding for a 253-amino-acid protein with a predicted 68-

residue N-terminal extension. Rtn1 and Rtn2 are closely related and share 91% amino acid identity for their longest predicted polypeptides (*Figure 1—figure supplement 2C*).

Reticulon proteins are known to oligomerize to induce ER membrane curvature (*Sparkes et al., 2010*; *Lee et al., 2013*; *Tolley et al., 2010*). Using a split-ubiquitin yeast-two hybrid (Y2H) assay, we found that maize Rtn1 (the longest polypeptide with a complete RHD domain) and Rtn2 interact with themselves and with each other (*Figure 1B*). We also expressed GFP fusions of Rtn1 or Rtn2 in maize mesophyll protoplasts together with the ER marker mCherry-KDEL, and found that both proteins localize to the ER, with a preference for tubular ER regions and junctions (*Figure 1C*). To test whether over-expression of maize Rtn1/2 re-models the ER, as seen with *Arabidopsis* RTNLB13 (*Tolley et al., 2008*), we expressed in *Arabidopsis* mesophyll protoplasts the ER lumenal marker CFP-KDEL together with increasing concentrations of a plasmid encoding Rtn2-mCherry under the control of the CaMV *35S* promoter. As shown in *Figure 1D and E*, transformation with higher concentrations of *35S:Rtn2-mCherry* increasingly altered the morphology of the ER and resulted in lower co-localization between CFP-KDEL and Rtn2-mCherry. This inverse correlation in signal colocalization is consistent with a reduced diffusion of ER lumenal contents due to Rtn-induced ER constrictions, as described previously for the over-expression of *Arabidopsis* RTNLB13 in plant cells (*Tolley et al., 2008*). Based on these results, we concluded that maize Rtn1/2 localize to the ER and are able to oligomerize and change the morphology of the ER, thus supporting their role as reticulon proteins.

## Rtn1 and Rtn2 interact with Atg8 through cytoplasmic and transmembrane AIMs

The RHD-containing proteins FAM134B and RTN3L in mammals and Atg40 in yeast act as selective autophagic receptors for ER turnover through macro-ER-phagy using their AIMs sequences to bind Atg8 (*Chiramel et al., 2016*; *Mochida et al., 2015*; *Khaminets et al., 2015*; *Bhaskara et al., 2019*). To determine whether maize Rtn1/2 similarly participate in ER-phagy, we tested whether they bind Atg8 using split ubiquitin Y2H assays; both Rtn1 and Rtn2 interacted with maize Atg8a (Zm00001d006474_P001), which is expressed in the developing maize endosperm at 18–22 days after pollination (*Li et al., 2015*; *Figure 2A*). These interactions were then confirmed by co-immunoprecipitation assays showing that HA-Rtn2 binds to GFP-Atg8a upon transient expression in *Arabidopsis* protoplasts (*Figure 2B*).

Assuming that Rtn1/2, like their mammalian and yeast counterparts, interact with Atg8a *via* one or more AIMs (*Figure 1A*), we scanned the proteins for AIM-type sequences using the iLIR autophagy database (*Jacomin et al., 2016*). Seven positionally-conserved AIMs were predicted in Rtn1/2, including one in the cytoplasmic loop, one in the C-terminal cytoplasmic tail, and five within the transmembrane regions (*Figures 1A* and *2C*; *Figure 1—figure supplement 2C*). To map the Atg8-interaction sites in Rtn2 by Y2H, we first tested the N- and C-terminal cytoplasmic tails as well as a fragment encompassing the cytoplasmic loop, and found that only the C-terminal region of Rtn2 interacted with Atg8a (*Figure 2D*), which is consistent with the presence of a single functional AIM at the C-terminus of both mammalian FAM134B and yeast Atg40 (*Khaminets et al., 2015*; *Mochida et al., 2015*). However, substitutions replacing the predicted C-terminal AIM with alanine residues in both Rtn1 and Rtn2 (Rtn1-mAIM1 and Rtn2-mAIM1) largely retained Atg8a binding in a split ubiquitin Y2H assay (*Figure 2E*; *Figure 2—figure supplement 1A*), prompting us to investigate other putative AIMs by both Y2H and co-immunoprecipitation. As shown in *Figure 2E–G*, only when alanine substitutions were collectively introduced into the predicted AIMs within the cytoplasmic loop (predicted AIM6), two other AIMs located at the transmembrane regions close to the cytoplasmic face of the ER membrane (predicted AIM2 and AIM5), and AIM1 within the C-terminal tail, did we successfully abolish Atg8a binding. Mutations in AIM6 at the cytoplasmic loop had a less pronounced effect on Atg8a binding (*Figure 2E–G*), suggesting that although each one of these four AIMs promotes the association of Rtn1/2 with Atg8a, their contributions are not equal. Importantly, these AIM mutations did not appear to alter the localization of Rtn2 in the ER membranes, as confirmed by their ability to interact with Rtn1-GFP in co-immunoprecipitation assays (*Figure 2—figure supplement 1B*) and to colocalize with the CFP-KDEL ER-marker (*Figure 2—figure supplement 1C*).

To determine whether the mutations in the AIMs located at the transmembrane regions affect Rtn2 protein stability, we detected both HA-Rtn2 and HA-Rtn2(mAIM1,2,5,6) by immunoblot assays upon inhibition of protein synthesis by cyclohexamide (CHX) in Arabidopsis protoplasts (*Figure 2H*).

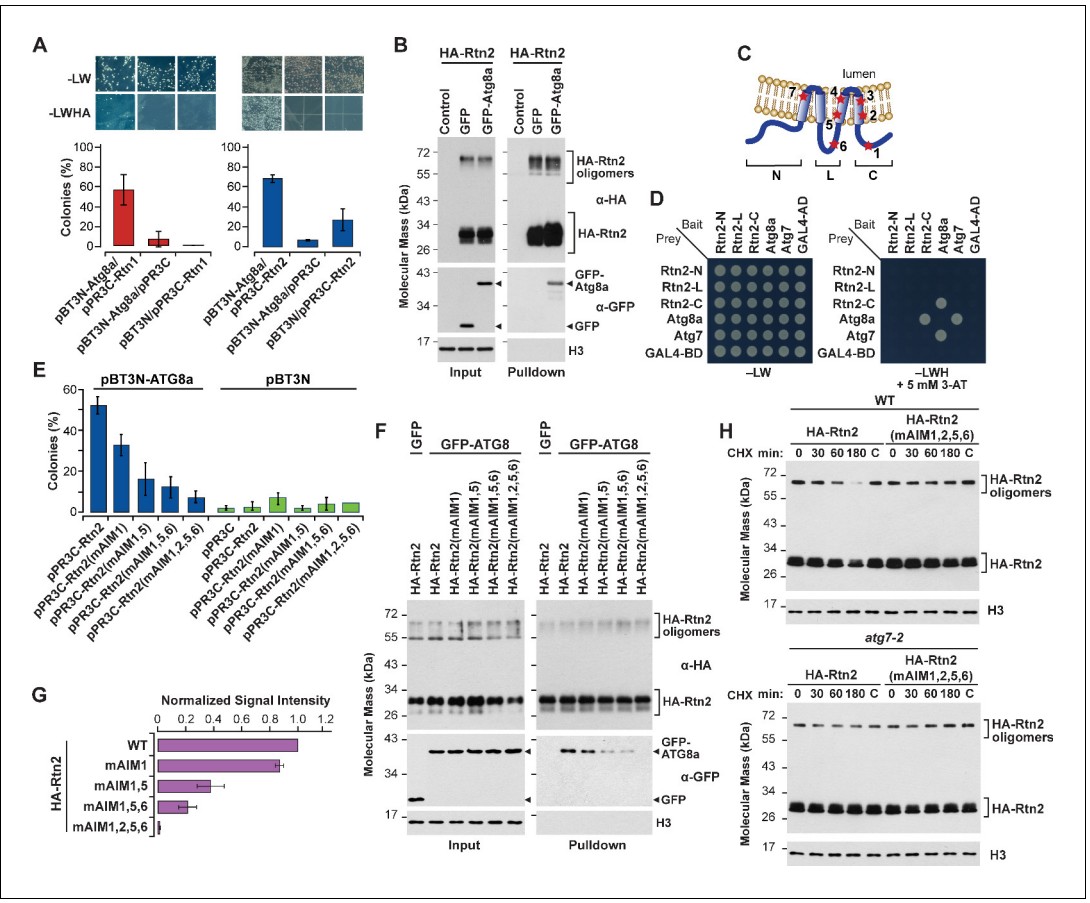

**Figure 2.** Rtn1 and Rtn2 interact with maize Atg8a. (**A**) Split-ubiquitin Y2H assays showing interactions between Rtn1/2 and Atg8a. Plates show colonies grown on −LW medium (selecting for transformation) and on −LWHA medium (selecting for interaction). Controls were performed by co-expressing Rtn1, Rtn2, or Atg8a proteins with the corresponding empty vector. Bar graphs show ratios between the number of colonies grown on −LWHA medium versus −LW medium. The ratios from control plates were subtracted from those testing direct interactions between Rtn proteins and Atg8a. Between 200 and 300 colonies were counted in each case. The graphs show the average of three independent experiments; error bars indicate SD. (**B**) Co-immunoprecipitation assays showing interactions between HA-Rtn2 and GFP-Atg8a. Immunoprecipitation was performed with anti-HA antibodies in *Arabidopsis* protoplasts transiently expressing HA-Rtn2 and either GFP alone (negative control) or GFP-Atg8a. Immunodetection of histone 3 (H3) was used to confirm near equal loading of the input fractions. (**C**) Diagram showing positions of the predicted AIMs in Rtn1 and Rtn2. (**D**) Y2H assays testing the interactions between Atg8a and either the N-terminal (N), cytoplasmic intermembrane loop (L), or C-terminal (C) regions of Rtn2. The interaction between Atg8a and Atg7 was used as a positive control. Negative controls included the individual prey or bait sequences expressed with the corresponding empty vectors. (**E**) Split-ubiquitin Y2H assay between Atg8a and Rtn2 or Rtn2 proteins with mutations in putative AIM domains. Bar graphs show ratios between the number of colonies grown on −LWHA medium versus −LW medium as in (**A**). (**F**) Co-immunoprecipitation of HA-Rtn2 or HA-Rtn2 mAIM mutant proteins with either GFP alone (negative control) or GFP-Atg8 in transformed *Arabidopsis* protoplasts. Immunoprecipitation was performed with anti-HA antibodies. Immunodetection of histone 3 (H3) was used to confirm near equal loading of the input fractions. (**G**) Densitometric quantification of GFP-Atg8 bands from the immunoblots shown in (**F**), plus two additional independent biological replicates. Bars represent the mean (± S.D.) normalized to the amount of GFP-Atg8a immunoprecipitated by wild-type HA-Rtn2. (**H**) Stability assay of HA-Rtn2 and HA-Rtn2(mAIM1,2,5,6) expressed in wild type (WT) or *atg7-2* mutant Arabidopsis protoplasts and treated with cycloheximide (CHX) for 0, 30, 60, and 180 min. Control protoplasts (C) not treated with CHX were collected and processed 180 min after the initiation of the CHX treatment.

The online version of this article includes the following figure supplement(s) for figure 2:

**Figure supplement 1.** Effects of mutations in Rtn AIM domains.

**Figure supplement 2.** Interactions between Rtn2 and Atg8a are promoted by ER stress-inducing agents but do not require Atg8 lipidation.

HA-mRtn2(AIM1,2,5,6) seemed to be more stable than HA-Rtn2 when expressed in wild-type Arabidopsis protoplasts treated with CHX for 180 min (*Figure 2H*), consistent with the inability of HA-mRtn2(AIM1,2,5,6) to be efficiently sequestered into autophagosomes for degradation. To further confirm that these AIMs are required for the autophagy-mediated degradation of Rtn2, we performed a similar CHX chase using Arabidopsis *atg7* mutant protoplasts, which are unable to perform macroautophagy (*Doelling et al., 2002*). In this situation, both HA-Rtn2 and HA-mRtn2(AIM1,2,5,6) showed similar stability, strongly implying that the degradation of HA-Rtn2 during the CHX treatment depends on macroautophagy (*Figure 2H*).

Since AIM2 and AIM5 are predicted to be at least partially embedded into the ER membrane, we speculated that Atg8a lipidation, and therefore incorporation of Atg8a into membranes, could influence Rtn1/2 binding. However, when we tested the G150A mutation of Atg8a (Atg8a-G150A), which prevents conjugation of Atg8a to phosphatidylethanolamine (*Li et al., 2015*), no impact on Atg8a binding to either HA-Rtn2 or HA-Rtn2-mAIM1 was seen as compared to that for wild-type Atg8a (*Figure 2—figure supplement 2A*). We also consistently detected interactions between GFP-Atg8a and HA-Rtn2 by co-immunoprecipitation assays using protoplasts from the Arabidopsis *atg5-1* mutant (*Figure 2—figure supplement 2B*), in which Atg8 lipidation is blocked (*Thompson et al., 2005*). Taken together, these results show that four AIMs in Rtn2, one located at the C-terminal region, one at the cytoplasmic loop, and two within the predicted transmembrane regions close to the cytoplasmic face of the ER membrane, contribute to its interaction with Atg8a *via* a mechanism that does not require Atg8 lipidation.

## Rtn1 and Rtn2 as autophagic receptors for ER turnover during ER stress

To test whether maize Rtn1 and Rtn2 are sequestered into autophagosomes, we triggered autophagic turnover of the ER in *Arabidopsis* protoplasts expressing GFP-Atg8a and Rtn2-mCherry using dithiothreitol (DTT) to induce ER stress (*Figure 3A–D*). After a 12 hr exposure to 2 mM DTT, we found Rtn2-mCherry signal associated with numerous cytoplasmic foci that were most likely autophagosomes since they were also decorated with GFP-Atg8a (*Figure 3C,D*). When we treated these protoplasts simultaneously with DTT and 1 μM Concanamycin A (ConA) to suppress vacuolar turnover (*Li et al., 2014*), we observed the accumulation of autophagic bodies containing both Rtn2-mCherry and GFP-Atg8a inside the central vacuole (*Figure 3E,F*), indicating that Rtn2 actively undergoes autophagy during ER stress.

Under the assumption that Rtn2 acts as an ER-phagy receptor, we tested whether its sequestration into autophagosomes depends on its interaction with Atg8, using several AIM mutants of Rtn2 mutants with dampened Atg8a binding. Here, we expressed mCherry fusions of Rtn2, Rtn2(mAIM1), and Rtn2(mAIM1,2,5,6) together with GFP-Atg8a in *Arabidopsis* protoplasts treated with DTT and ConA. Autophagic bodies were observed in all three cases but at different frequencies. Whereas on average 54% of the protoplasts contained autophagic bodies labeled with both GFP-Atg8a and Rtn2-mCherry, this frequency was reduced to 42% and 23% for protoplasts expressing Rtn2(mAIM1)-mCherry and Rtn2(mAIM1,2,5,6)-mCherry, respectively (*Figure 3E–K*). A likely explanation for the low but detectable recruitment of Rtn2(mAIM1,2,5,6)-mCherry into autophagosomes is its possible interaction with endogenous Arabidopsis Rtn proteins that could themselves act as ER-phagy receptors.

We also tested whether Rtn2 redistributes within the ER in response to ER stress by examining the localization of Rtn2-mCherry and CFP-KDEL in protoplasts incubated for 3 hr with or without DTT. We did not detect changes in the mCherry signal relative to that of CFP after DTT treatment (*Figure 4A*), suggesting that Rtn2 does not redistribute within the ER in response to protein folding stress.

To assay whether the affinity of Rtn2 for Atg8a was enhanced under ER-stress, we incubated protoplasts expressing GFP-Atg8a and HA-Rtn2 for 3 hr in the presence or absence of 2 mM DTT, and then co-immunoprecipitated GFP-Atg8a with anti-HA antibodies. An approximately 3-fold increase in the binding between GFP-Atg8a and HA-Rtn2 was seen upon DTT treatment (*Figure 4B–D*; *Figure 2—figure supplement 2A*). Similar results were observed when we performed a reciprocal co-immunoprecipitation of HA-Rtn2 with anti-GFP antibodies (*Figure 2—figure supplement 2C*). We also tested the effect of tunicamycin, which induces ER stress by impairing N-glycosylation of nascent proteins inside the ER. Similar to DTT, exposing *Arabidopsis* protoplasts to 5 μg/mL

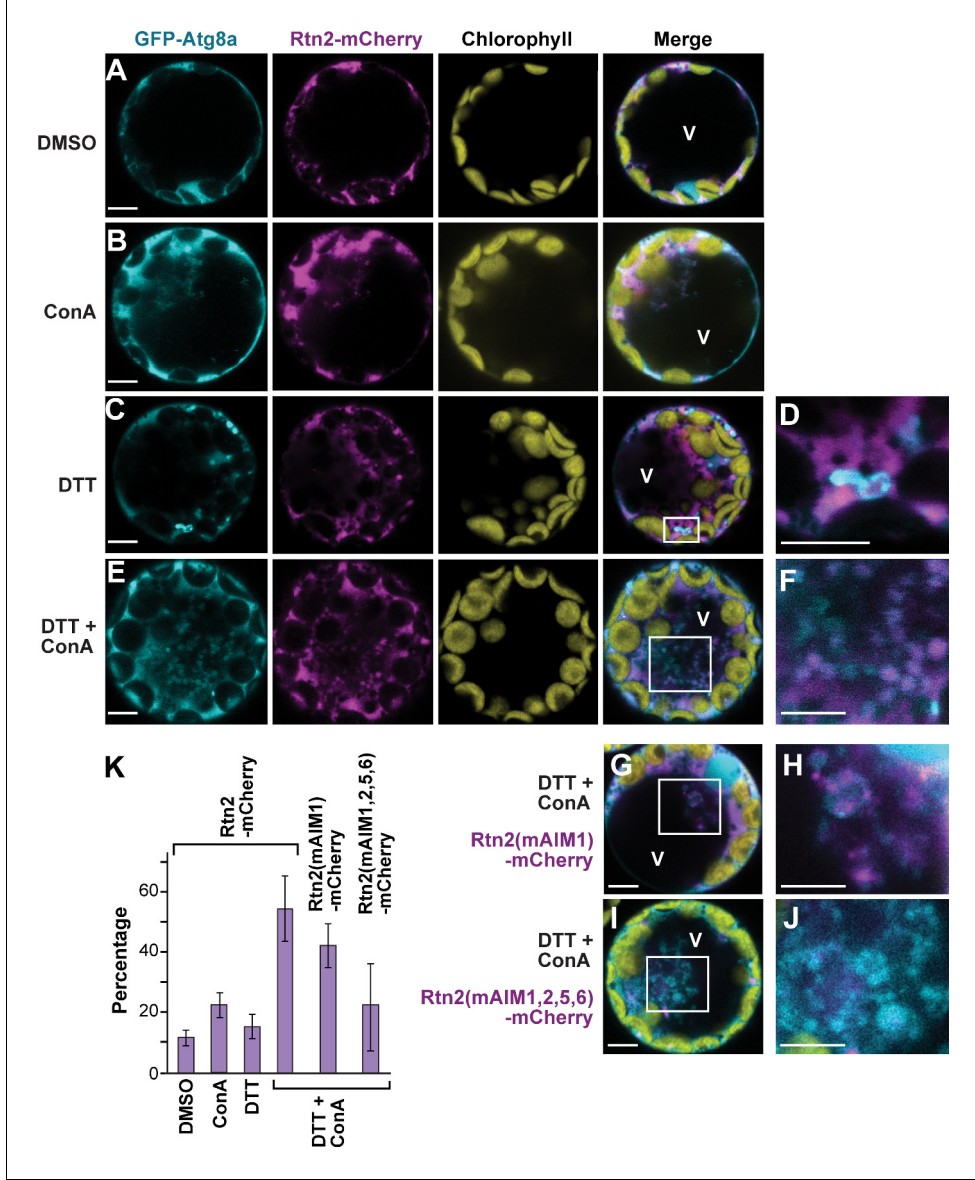

**Figure 3.** ER stress promotes association of Rtn2 with autophagic organelles during ER-phagy. (A–J) Co-expression of GFP-Atg8a with either Rtn2-mCherry (A–F), Rtn2(mAIM1)-mCherry (G, H), or Rtn2(mAIM1,2,5,6)-mCherry (I, J) in *Arabidopsis* protoplasts treated with 1 µM ConA, 2 mM DTT, and/or DMSO (control). Shown are representative protoplasts imaged by confocal fluorescence microscopy. (D), (F), (H), and (J) are enlarged images of the areas highlighted by white boxes in the preceding merged channel images. (D) shows a putative autophagosome decorated with GFP-Atg8a and containing Rtn2-mCherry. (F), (H) and, (J) show autophagic bodies inside the central vacuole (V). Note that autophagic bodies in (J) are positive for GFP-Atg8a but largely devoid of Rtn2(mAIM1,2,5,6)-mCherry. (K) Percentage of observed protoplasts showing vacuolar autophagic bodies positive for both GFP-Atg8a and Rtn2-mCherry. Graph shows average of three independent biological replicates (± S.D.). Between 10 and 15 randomly selected protoplasts were analyzed in each case. Scale bars = 5 µm.

tunicamycin for 3 hr led to increased binding between GFP-Atg8a and HA-Rtn2 (*Figure 2—figure supplement 2D,E*).

To identify which AIMs in Rtn2 might be responsible for the enhanced binding to Atg8a under ER stress, we performed co-immunoprecipitation assays with GFP-ATG8a and HA-tagged versions of either Rtn2, Rtn2(mAIM1), or Rtn2(mAIM1,2,5,6) expressed in protoplasts treated with or without DTT. As shown in *Figure 4B and C*, no increase in Atg8a binding was seen for the Rtn2(mAIM1) protein, indicating that AIM1 within the C-terminal tail of Rtn2 was required for the increased binding of

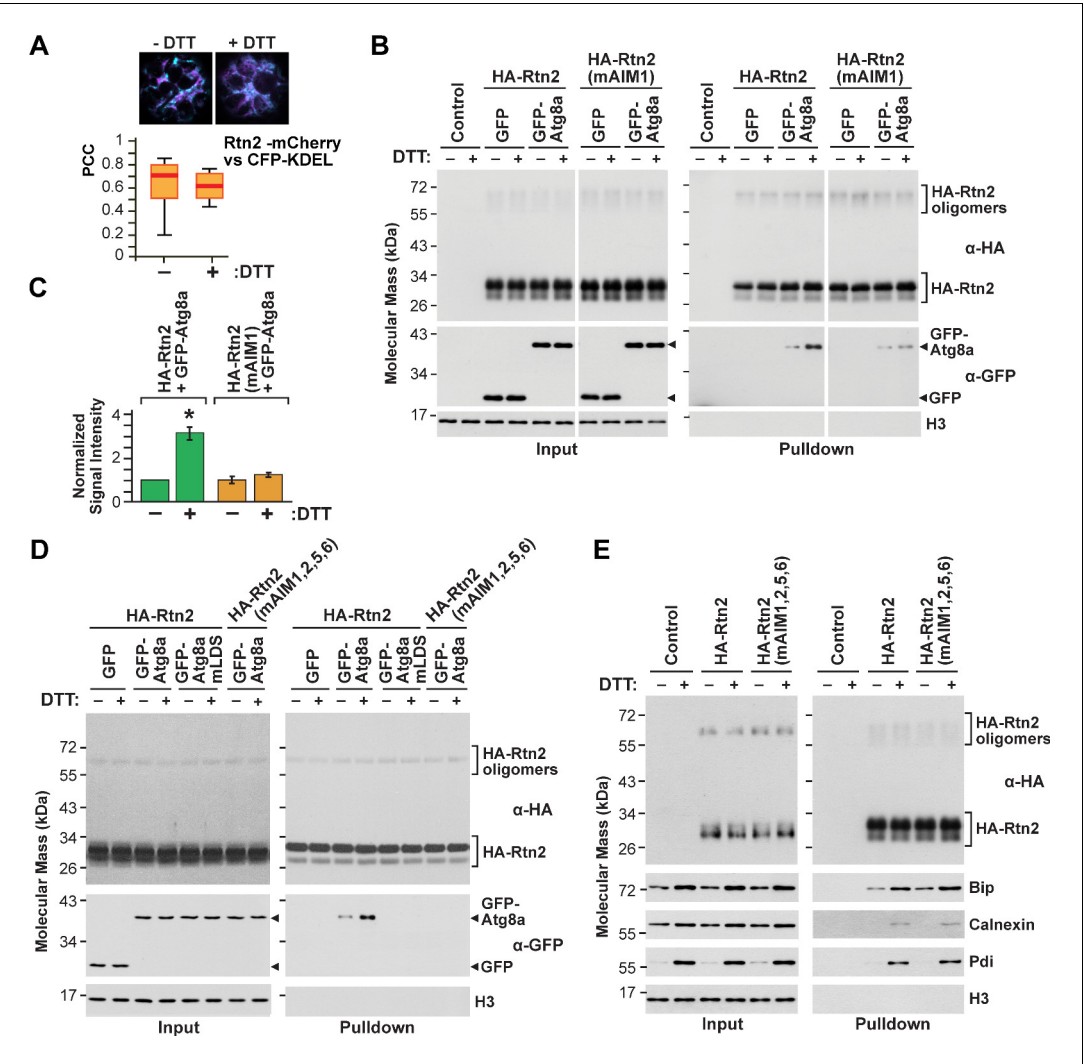

**Figure 4.** Rtn2 distribution and interactions during ER stress. (A) Colocalization analysis of Rtn2-mCherry and CFP-KDEL in protoplasts treated with or without DTT for 3 hr. One representative protoplast of each treatment is shown. The box and whisker plot represents the Pearson correlation coefficients (PCC) calculated for Rtn2-mCherry and CFP-KDEL. The experiment was performed three times; graph shows the results of a representative experiment. At least 10 randomly selected protoplasts for each treatment were included in the analysis. (B) Co-immunoprecipitation of HA-Rtn2 or HA-Rtn2(mAIM1) mutant protein with either GFP alone (negative control) or GFP-Atg8 from transformed *Arabidopsis* protoplasts treated with or without DTT for 3 hr. (C) Densitometric quantification of GFP-Atg8 bands from the immunoblots shown in B), plus two additional independent biological replicates. Bars represent the mean (± S.D.) normalized to the amount of GFP-Atg8a immunoprecipitated by wild-type HA-Rtn2 in the absence of DTT treatment. The asterisk indicates a significant difference between GFP-Atg8 bands from protoplasts treated with or without DTT as determined by one-way ANOVA followed by Tukey's post-hoc tests, p<0.01. Note the increase in GFP-Atg8 bound to HA-Rtn2 but not to HA-Rtn2(mAIM1) in samples treated with DTT. (D) Co-immunoprecipitation of HA-Rtn2 or HA-Rtn2(mAIM1,2,5,6) with GFP alone (negative control), GFP-Atg8, or GFP-Atg8a(mLDS) from transformed *Arabidopsis* protoplasts treated with or without DTT for 3 hr. (E) Co-immunoprecipitation of HA-Rtn2 with calnexin, Bip, and Pdi from protoplasts treated with or without DTT, as in (B). In (B), (D), and (E), the immunoprecipitation was performed with anti-HA antibodies and immunodetection of histone 3 (H3) was used to confirm near equal loading for the input fractions.

The online version of this article includes the following figure supplement(s) for figure 4:

**Figure supplement 1.** Interactions between Rtn2 and Atg8a are not affected by the cysteine residues near the hinge region of Rtn2.

**Figure supplement 2.** Co-immunoprecipitation assay of HA-Rtn2 and ER proteins.

Rtn2 to Atg8a upon DTT exposure (*Figure 4B,C*). Under similar co-immunoprecipitation conditions, the HA-Rtn2(mAIM1,2,5,6) mutant protein did not appreciably interact with GFP-Atg8a, either in the presence or absence of DTT, again confirming that Rtn2 employs several AIMs to bind Atg8. To confirm that the interaction between Atg8a and Rtn2 after DTT treatment was solely dependent on AIM-mediating binding, we introduced two point mutations (Y50A, L51A) in the LIR/AIM docking site (LDS) of maize Atg8a that should abolish AIM binding (*Marshall et al., 2019*). Accordingly, GFP-Atg8a(mLDS) did not interact with HA-Rtn2 in protoplasts incubated with or without DTT (*Figure 4D*).

DTT is a reducing agent that causes protein misfolding in the ER by breaking disulfide bridges between cysteines. We noticed that both Rtn1 and Rtn2 contain four positionally conserved cysteines within the N-terminal and the C-terminal hairpin transmembrane segments of the RHD that are predicted to form hinges near the ER lumen (C107, 109, 190, and 191; *Figure 4—figure supplement 1A*). Based on the topology of *Arabidopsis* RTNLB proteins (*Sparkes et al., 2010*) and modeling predictions of maize Rtn1 and Rtn2 by TOPCONS (http://topcons.cbr.su.se), these cysteine-rich regions could be fully or partially exposed to the ER lumen, and thus could act as 'sensors' of oxidative stress in the ER by enhanced binding of Rtn1/2 to Atg8a after DTT treatment. To test this notion, we replaced the four cysteines in Rtn2 with alanines and assayed for altered interactions between this HA-Rtn2(CA) mutant and GFP-Atg8a in protoplasts. However, HA-Rtn2 and HA-Rtn2(CA) bound Atg8 at similar levels with or without DTT treatment (*Figure 4—figure supplement 1B*). Likewise, we found that lipidation of Atg8a was not required for its increased binding to HA-Rtn2 upon DTT treatment, as demonstrated by co-immunoprecipitation assays between HA-Rtn2 and GFP-Atg8a(G150A) (*Figure 2—figure supplement 2A*).

Recently, the ER-phagy receptor FAM134B was shown to interact with the ER-resident transmembrane chaperone calnexin, which is part of the ER protein folding machinery (*Forrester et al., 2019*). This interaction occurs through the transmembrane region of FAM134B and is needed for disposing misfolded proteins such as procollagen *via* ER-phagy. However, FAM134B binding to calnexin was reported to remain unchanged during misfolded protein accumulation (*Forrester et al., 2019*). To test whether Rtn2 similarly interacts with the maize ER protein folding machinery as a potential sensing mechanism for recruiting Atg8a during ER stress conditions, we tested for the ability of Rtn2 to bind calnexin and two ER-lumen soluble proteins involved in protein folding, Bip and Protein disulfide isomerase (Pdi). As shown in *Figure 4E*, HA-Rtn2 not only co-immunoprecipitated all three ER proteins, but this binding was increased upon DTT treatment. While the change in the abundance of Bip, calnexin, and Pdi between DTT-treated and untreated protoplasts made it difficult to estimate changes in their binding affinity for Rtn2, their increased abundance under ER stress nevertheless resulted in a higher association. Other ER-localized proteins, such as the ER calcium channel ACA2 (*Harper et al., 1998*) or the ER lumen marker CFP-KDEL, failed to co-immunoprecipitate with HA-Rtn2 in either DTT-treated or untreated Arabidopsis protoplasts (*Figure 4—figure supplement 2*), suggesting that the association of Rtn2 and the ER chaperones is specific.

## Characterization of *rtn1* and *rtn2* mutants in maize

To examine the functions of Rtn1 and Rtn2 in maize, we identified a collection of mutants generated by *UniformMu* transposition that were predicted to impact expression of both genes. One *Mu* insertion in *Rtn1* disrupted the coding region within the first exon (*rtn1-1*), whereas three *Mu* insertions for *Rtn2* disrupted the coding region within either the first or second exons (*rtn2-1*, *rtn2-2*, and *rtn2-3*), and thus all were predicted to be strong alleles (*Figure 5A,B*). We confirmed by RT-PCR that all four lines produced aberrant transcripts directing the synthesis of abnormal proteins (*Figure 5A,B*; *Figure 5—figure supplement 1A,B*). The *rtn1-1* transcript sequence would translate into a shorter Rtn1 polypeptide with no transmembrane segments (*Figure 5A*; *Figure 5—figure supplement 1A*), both the *rtn2-1* and *rtn2-3* insertions introduced early stop codons, and the *rtn2-2* insertion introduced a 6-amino-acid deletion/3-amino-acid insertion near the N-terminus (*Figure 5B*; *Figure 5—figure supplement 1B*). All four homozygous single mutants grew normally and generated viable progeny, indicating that neither Rtn1 nor Rtn2 are essential for maize vegetative and seed development by themselves (*Figure 5—figure supplement 2A,B*).

As ER re-modeling is important for storing nutrients within the maize endosperm, we tested whether the accumulation of storage proteins in ER protein bodies was impacted in the *rtn1/2* lines. Upon microscopic analysis of the starchy endosperm cells, it was evident that the *rtn1* and *rtn2*

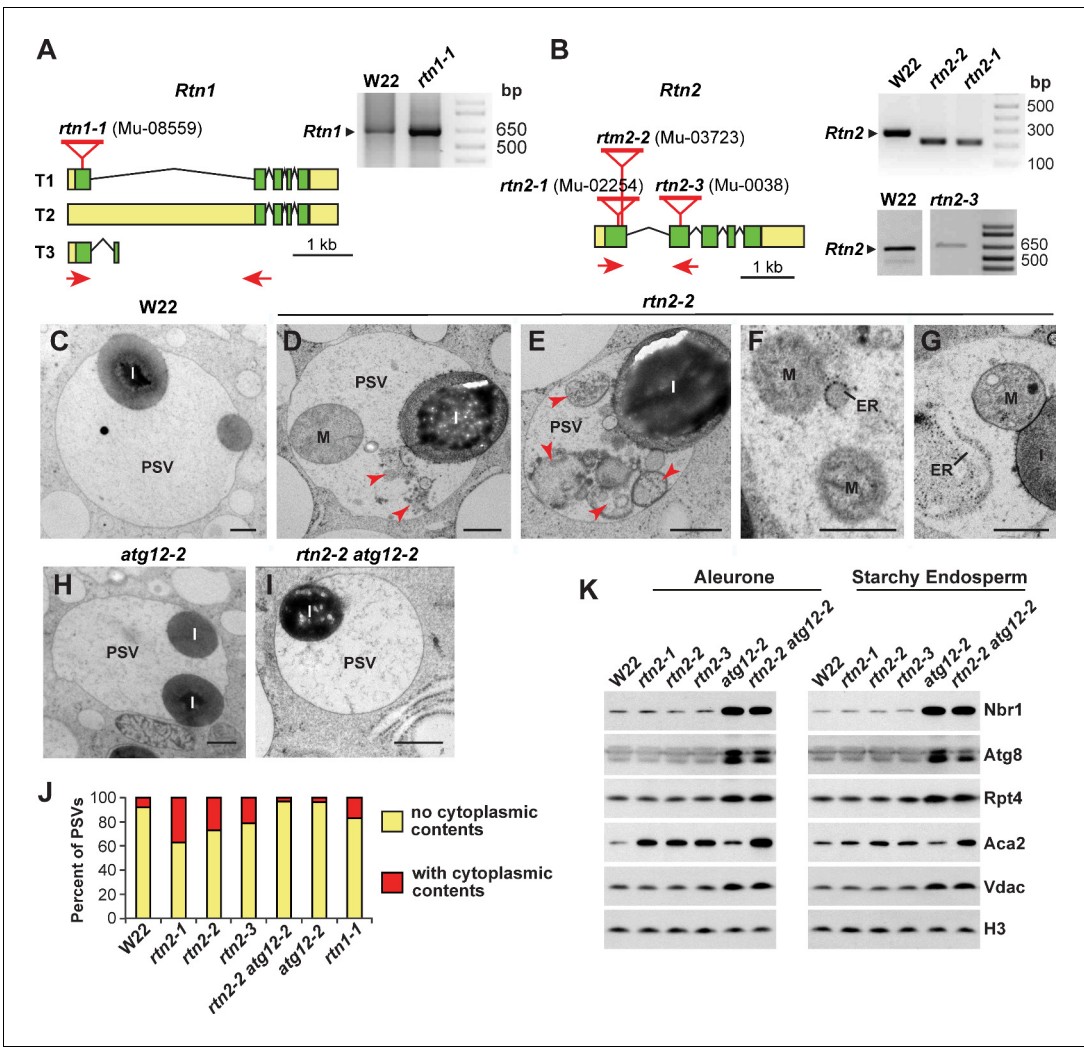

**Figure 5.** Characterization of maize *rtn1* and *rtn2* mutants. (**A, B**) Diagrams of *Rtn1* (**A**) and *Rtn2* (**B**) showing the position of the *Mu* insertions, plus RT-PCR of the mutated transcripts (see *Figure 5—figure supplement 1* for predicted protein sequences derived from the amplified transcripts). (**C–I**) Transmission electron micrographs of protein storage vacuoles (PSVs) in aleurone cells from homozygous wild-type W22 (**C**), *rtn2-2* (**D–G**), *atg12-2* (**H**), and *rtn2-2 atg12-2* (**I**) developing seeds at 20 days after pollination. Note the accumulation of ER, whole mitochondria (M), and other membranous structures (red arrowheads) in the vacuolar matrix of the *rtn* mutants. (**J**) Percentage of vacuoles showing cytoplasmic contents in the vacuolar matrix. At least 100 PSVs of each genotype were used for the analysis. (**K**) Immunoblot analysis of autophagy related proteins (Nbr1 and Atg8), the proteasome component Rpt4, the ER-resident channel Aca2, and the mitochondrial protein Vdac in aleurone and starchy endosperm samples from wild-type W22, the three *rtn2* mutants, *atg12-2*, and *rtn2-2 atg12-2* developing seeds. Immunodetection of histone 3 (H3) was used to confirm near equal protein loading. Scale bars = 500 nm. The online version of this article includes the following figure supplement(s) for figure 5:

**Figure supplement 1.** Predicted amino acid sequences of Rtn1 and Rtn2 polypeptides.
**Figure supplement 2.** Plant growth and seed development for the maize *rtn1* and *rtn2* mutants.
**Figure supplement 3.** Abnormal contents in protein storage vacuoles (PSVs) of *rtn1/2* mutant aleurone cells.

single mutants accumulated normal protein bodies and that they had similar storage protein profiles as compared to their wild-type W22 parent (*Figure 5—figure supplement 2C,D*). We also analyzed lipid droplet morphology in the developing aleurone cells by staining neutral lipids with Nile blue and by transmission electron microscopy. All the *rtn1/2* mutants formed morphologically normal lipid droplets (*Figure 5—figure supplement 2E–G*). We then visualized aleurone protein storage vacuoles to assess vacuolar delivery and found that the *rtn1/2* mutants formed morphologically

normal storage protein inclusions. However, while such aleurone vacuoles from W22 seeds were typically empty of other inclusions besides the protein accretions, the vacuoles from *rtn1* and *rtn2* aleurone cells were frequently filled with cytoplasmic material, including ER, mitochondria, and other organelles (*Figure 5C–G*; *Figure 5—figure supplement 3*), suggesting mis-regulated autophagy.

## Rtn1 and Rtn2 regulate macroautophagy

To further connect the maize Rtn1/2 proteins with autophagy, we introgressed the *rtn2-2* allele into the previously described *atg12-2* mutant, which poorly lipidates ATG8 and is thus attenuated in macro-autophagy (*Li et al., 2015*). The elevated accumulation of cytoplasmic material inside aleurone vacuoles was no longer evident in *rtn2-2 atg12-2* plants (*Figure 5H–J*), indicating that the loss of Rtn2 leads to an increase in Atg8-mediated macro-autophagy.

Plants compromised in autophagy typically have elevated levels of autophagic components and cargo receptors, given their impaired breakdown (*McLoughlin et al., 2018*; *Jung et al., 2020*). Consistent with most aspects of autophagy being normal in the *rtn2* mutant lines, we failed to see increases in the abundance of Atg8 and the autophagy receptor Nbr1 in either aleurone or starchy endosperm cells from the *rtn2-1, rtn2-2, and rtn2-3* plants, while both proteins hyper-accumulated in endosperm samples from the *atg12-2 and rtn2-2 atg12-2* plants (*Figure 5K*). We then analyzed the accumulation pattern of potential autophagic cargo from several cellular compartments. The mitochondrial protein Vdac (Voltage-dependent anion channel) and the proteasome component Rpt4, both targets of autophagy (*Spitzer et al., 2015*; *Marshall et al., 2015*; *Li et al., 2015*), only showed hyperaccumulation in endosperm samples containing the *atg12-2* mutation (*Figure 5K*). In contrast, the ER calcium channel Aca2 hyper-accumulated in the *rtn2-1, rtn2-2, and rtn2-3* aleurone and starchy endosperm, as well as in the *atg12-2*, and *rtn2-2 atg12-2* endosperm samples, suggesting that whereas mutations in *atg12* lead to the general accumulation of autophagy cargo from different cellular compartments, the *rtn2* mutations more specifically impact the ER. This hyper-accumulation of Aca2 in the *rtn2* mutants was more obvious in aleurone versus starchy endosperm cells, implying that the aleurone is particularly sensitive to the loss of Rtn2 (*Figure 5K*).

To confirm the mis-regulation of autophagy in *rtn1/2* aleurone cells seen by transmission electron microscopy, we introgressed the *rtn2-3* mutation into maize wild type (W22) and *atg12* mutant plants expressing the YFP-Atg8a reporter, which labels autophagosomes upon lipidation and is delivered to the vacuole as part of the autophagic body (*Li et al., 2015*). As expected, the YFP-Atg8a signal in wild-type aleurone cells localized to both the cytosol and cytoplasmic puncta of less than 1 μm in diameter, which likely represented autophagosomes (*Figure 6A,B*; red arrowheads), but remained exclusively cytosolic when either the *atg12-1* or *atg12-2* (*Li et al., 2015*) mutations are introduced (*Figure 6C,D,H,I*). However, in *YFP-Atg8a rtn2-3* aleurone cells, a strong YFP signal was detected inside the vacuolar matrix (*Figure 6E–G*), in agreement with the abnormally high delivery of autophagy cargo observed by electron microscopy (*Figure 5D–G*; *Figure 5—figure supplement 3*). Consistent with our electron microscopy analysis, this fluorescent vacuolar signal was absent in aleurone cells from YFP-Atg8a *rtn2-3 atg12-2* endosperm, which are deficient in macro-autophagy (*Figure 6H,I*).

We next analyzed autophagic flux in aleurone and starchy endosperm cells by measuring the release of YFP from the YFP-Atg8a reporter. When proteins with GFP-based fluorescent tags are delivered to vacuoles, the GFP/YFP moiety is quickly released by vacuolar proteases and stably accumulates in the vacuolar lumen, which can then be assayed by immunoblot analysis of total cell lysates with anti-GFP antibodies (*Li et al., 2015*). Thus, the ratio of free YFP to YFP-Atg8a informs on the extent of vacuolar YFP-Atg8 delivery and can be used as a proxy for measuring overall autophagic flux. As shown in *Figure 6J and K*, the relative abundance of free YFP was higher in the *rtn2-3* aleurone samples compared to those from wild type, but was similar in the two starchy endosperm samples (*Figure 6J,K*), indicating that autophagic flux was specifically elevated in aleurone cells of the *rtn2-3* mutant. As expected, free YFP failed to accumulate in *rtn2-3 atg12-2* double mutant samples, confirming that the observed release of YFP from YFP-Atg8 was autophagy-dependent.

Our studies in protoplasts showed that: i) increased *Rtn2* expression can re-model the ER (*Figure 1C*); and ii) Rtn2 and Atg8 binding increases upon DTT or tunicamycin treatment (*Figure 4B, C*; *Figure 2—figure supplement 2C,D*). We therefore asked whether ER stress induces *Rtn1* and/or *Rtn2* expression in aleurone cells as a means to facilitate ER re-modeling and/or increase the pool of Rtn molecules needed to drive ER-phagy. Here, we used quantitative real-time (qRT)-PCR to

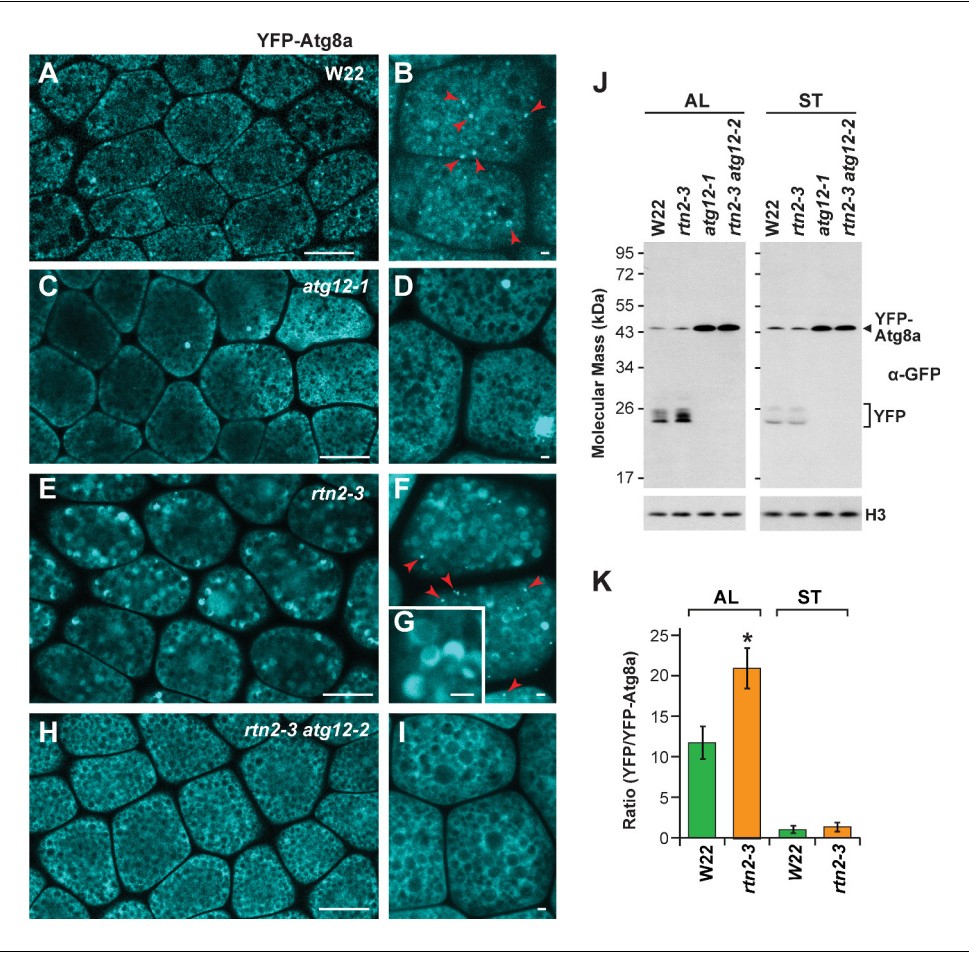

**Figure 6.** Autophagic flux is increased in *rtn2* mutant aleurone cells. (**A–I**) Confocal imaging of YFP-Atg8a in W22 (**A, B**), *atg12-1* (**C, D**), *rtn2-3* (**E, G**), and *rtn2-3 atg12-2* (**H, I**) aleurone cells from seeds harvested at 20 days after pollination. YFP-Atg8a-positive puncta consistent with autophagosomes were detected in W22 (**B**) red arrowheads) and *rtn2-3* (**F**), red arrowheads) aleurone cells but not in lines harbouring either the *atg12-1 or atg12-2* mutations. Notice the strong, crescent-shape YFP signal detected in the matrix of *rtn2-3* aleurone vacuoles in (**G**). (**J**) Immunoblot detection of YFP-Atg8a and cleaved YFP with anti-GFP/YFP antibodies. Immunodetection of histone 3 (H3) was used to confirm near equal protein loading. (**K**) Densitometric quantification of the YFP/YFP-Atg8a ratio from the immunoblots shown in (j), plus two additional independent biological replicates. Bars represent the mean (± S.D.). The asterisk indicates a significant difference between the *rtn2-3* mutant and the wild type as determined by one-way ANOVA followed by Tukey's post-hoc tests, (p<0.05). Note the increased release of YFP from YFP-Atg8a in aleurone but not starchy endosperm cells of the *rtn2-3* mutant. Scale bars = 20 µm in (**A**), (**C**), (**E**), (**H**), and 1 µm in (**B**), (**D**), (**F**), (**G**), (**I**).

measure the expression of *Rtn1* and *Rtn2* in aleurone cells incubated for 3 hr with or without DTT. Whereas a robust 2.7-fold increase was seen within 3 hr of DTT treatment for the *Bip2* transcript encoding an ER chaperone responsive to ER stress (*Wang et al., 2014*; *Kirst et al., 2005*), no change was seen for the *Rtn1* or *Rtn2* transcripts (*Figure 7A*).

## Rtn proteins help repress ER stress

The increased overall autophagic flux in *rtn1* and *rtn2* mutant aleurone cells is consistent with an inhibitory role of maize Rtn proteins in autophagy, which is counterintuitive to their role as selective ER-phagy receptors. A possible explanation is that mutations in Rtn proteins such as Rtn1/2 affect ER homeostasis and hence cause ER stress, which is well known to promote both ER-phagy and general autophagy in a variety of organisms and cell systems (*Yang et al., 2016*; *Ogata et al., 2006*).

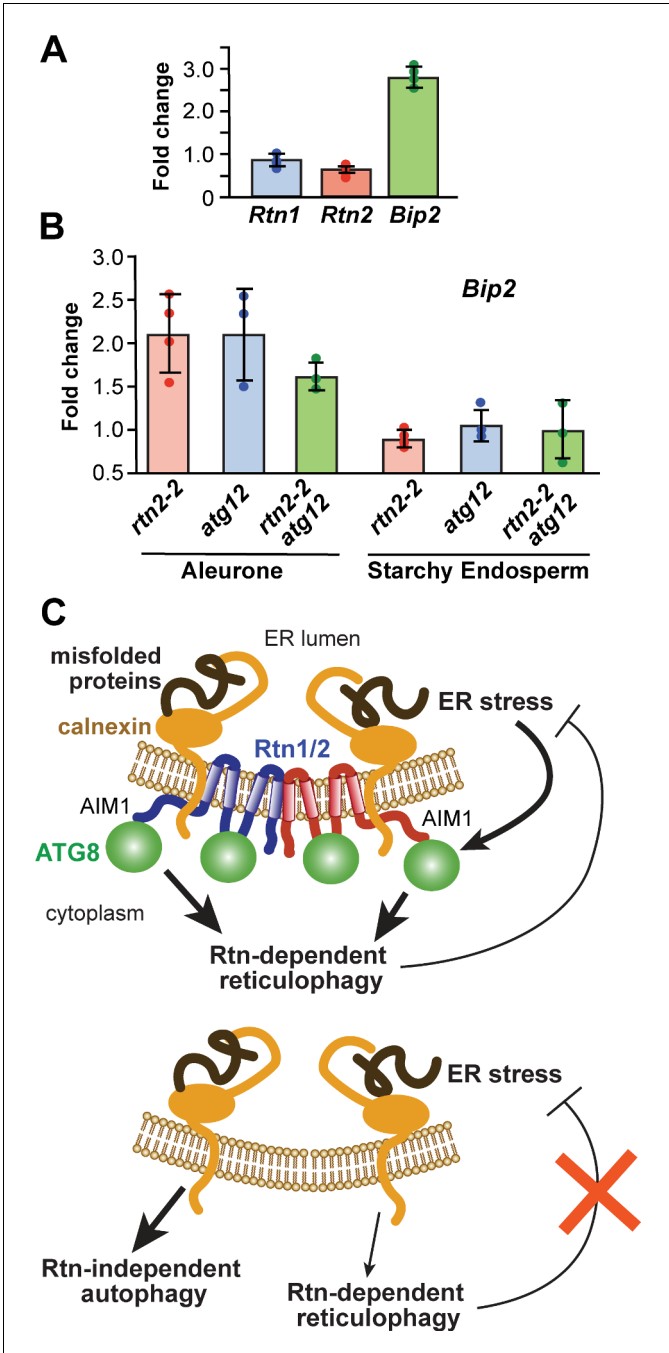

**Figure 7.** ER-stress in *rtn2* and *atg12* mutant endosperm cells. (**A**) qRT-PCR of *Rtn*1 and *Rtn*2 from aleurone cells treated with 2 mM DTT for 3 hr. Amplification of *Bip2* transcripts was used as an indicator of UPR induction. The graph combined results from four independent biological replicates, each consisting of three technical replicates for samples treated either with or without DTT. (**B**) qRT-PCR quantification of *Bip2* transcripts in W22, *rtn2-2*, *atg12-2*, and *rtn2-2 atg12-2* aleurone and starchy endosperm cells. *EF1α* was used as reference gene in (**A**) and (**B**). (**C**) Model of the action of Rtn proteins in maize aleurone cells and the response of *rtn1/2* mutant cells. When Rtn-mediated ER-phagy and/or other Rtn functions are reduced in aleurone cells, ER stress increases, promoting general autophagy as a compensatory mechanism for ER turnover.

Therefore, in the absence of Rtn-mediated ER-phagy, bulk autophagy might attempt to compensate for impaired ER turnover.

To examine whether the loss of *Rtn2* induces ER stress in developing (20 days after pollination) aleurone and starchy endosperm cells, we again measured mRNA levels for *Bip2* and found an approximately a 2-fold increase in *rtn2-2* aleurone samples as compared to the wild-type W22 (*Figure 7B*), an increase comparable to that measured in wild-type aleurone samples exposed to DTT (*Figure 7A*). Consistent with other results in this study (*e.g.* autophagy flux measurements and electron microscopy), the *rtn2-2* starchy endosperm failed to show this hyper-accumulation of the *Bip2* transcript, suggesting that Rtn2 is more influential in repressing ER stress in aleurone cells relative to the starchy endosperm. Interestingly, the *atg12-2* mutant also showed a 2-fold increase in the accumulation of *Bip2* transcripts in aleurone but not in starchy endosperm cells, though the combination of both *atg12-2* and *rtn2-2* mutations did not lead to a further increase in *Bip2* transcript levels in aleurone cells (*Figure 7B*). Collectively, these results indicate that both Rtn2 and a functional autophagy pathway are required to alleviate ER stress in aleurone cells.

## Discussion

Both the importance of reticulons to ER homeostasis in general and seed development in particular, and the participation of specific reticulon isoforms from other organisms in ER-phagy, promoted us to investigate the roles of Rtn1 and Rtn2 in maize seed development. Given their high expression levels in both the maize starchy endosperm and aleurone tissues, we hypothesized they could strongly influence ER functions, especially during the accumulation of lipid droplets and protein bodies. In this study, we showed that these two reticulon proteins help promote ER homeostasis in the aleurone cells of the maize endosperm. We also found that Rtn1 and Rtn2 act as ER-phagy receptors in plant cells under ER stress by binding to Atg8 *via* one or a combination of four AIMs, one at the C-terminal region, one at the cytoplasmic loop, and two within the predicted transmembrane regions close to the cytoplasmic face of the ER membrane. Why Rtn1/2 harbor multiple AIMs is unclear but their presence could help these reticulons maintain contact with Atg8-decorated autophagic membranes as the ER pinches off in small fragments during ER-phagy.

At least for Rtn2, ER stress increases its association with Atg8a, likely through increased affinity of the C-terminal AIM1 with the LDS binding pocket in Atg8. Our studies on *rtn2* mutants revealed that Rtn2 is not essential for maize growth and development, but critical for ameliorating ER stress in the aleurone cells, likely through their role(s) in ER-phagy. Loss of Rtn2 instead leads to elevated bulk autophagy likely as a compensatory mechanism when ER-phagy is impaired (*Figure 7C*).

### Involvement of reticulons in ER-phagy

ER-phagy is essential for cell survival, especially under conditions that trigger ER stress caused by the accumulation of misfolded proteins (*Yang et al., 2016*; *Pérez-Martín et al., 2014*; *Ogata et al., 2006*; *Khaminets et al., 2015*). In animals, six different ER-resident proteins (ATL3, CCPG1, FAM134B, RTN3L, SEC62, and TEX264) with different localization and expression patterns are known to act as ER-phagy receptors (*Chen et al., 2019*; *An et al., 2019*; *Khaminets et al., 2015*; *Grumati et al., 2017*; *Fumagalli et al., 2016*; *Smith et al., 2018*; *Chino et al., 2019*), consistent with a complex spatial and temporal cargo selection pattern. In response to starvation, RTN3L and ATL3 mediate constitutive clearance of ER tubules, whereas FAM134B removes ER sheets (*Khaminets et al., 2015*; *Grumati et al., 2017*). CCPG1 localizes to perinuclear and cortical ER, is up-regulated in response to ER stress, and protects against ER lumenal protein aggregation (*Smith et al., 2018*), while SEC62 is part of the translocon complex and regulates ER turnover after transient ER stress as a mechanism to adjust ER volume (*Fumagalli et al., 2016*). TEX264 is required for ER-phagy under both basal and starvation conditions and is expressed more ubiquitously than the other mammalian ER-phagy receptors (*Chino et al., 2019*; *An et al., 2019*).

All six mammalian ER-phagy receptors are tethered to the ER membrane through one or more transmembrane regions. It has been hypothesized that the known ER-phagy receptors with one or two transmembrane domains (*e.g.* CCPG1, SEC62, and TEX264) act early as linkers connecting the ER with autophagic membranes, whereas the RHD-containing receptors with multiple transmembrane segments participate at later stages in re-modeling and/or fragmenting ER domains for their sequestration into autophagosomes (*Wilkinson, 2020*). In fact, some reticulons work together with

TEX264, which has been postulated to act as a master receptor for ER-phagy (*Chen et al., 2019*). However, no obvious TEX264 homologues can be found outside vertebrates, suggesting that the regulation and mechanistic action of ER-phagy receptors differ significantly among taxa.

The six mammalian ER-phagy receptors contain one (CCPG1, FAM134B, and SEC62), two (ATL3), or six (RTN3L) AIMs/LIRs or GIMs that associate with Atg8/LC3 or the Atg8-type protein GABARAP, respectively. In all these cases, the AIM/LIR/GIM sequences are located within the cytoplasmic regions of the receptors, with the six AIMs of RTN3L found near the N-terminus and the single AIM of FAM134B within the C-terminal tail. We identified four functional AIMs in maize Rtn1/2, one located within the C-terminal tail similar to FAM134B, one in the cytoplasmic loop, and two additional AIMs in transmembrane regions. The presence of two functional AIMs at the transmembrane segments is puzzling. However, these AIMs are predicted to be near the cytoplasmic face of the ER membrane and therefore, could be partially accessible to Atg8 for binding. In addition, reticulon proteins are known to induce membrane stretching, which could increase the accessibility of transmembrane domains to cytosolic proteins. A recent study of FAM134B shows that the two wedge-shaped transmembrane helical hairpins of the RHD domain tend to create membrane asymmetry by stretching the cytoplasmic leaflet of the ER (*Bhaskara et al., 2019*). This membrane deformation is enhanced by the oligomerization of FAM134B mediated by interactions through the transmembrane domains. One could envision that membrane stretching on the cytoplasmic leaflet of the ER membrane mediated by the RHD domain induces a change in the packing of phospholipids around the transmembrane domains, increasing access of Atg8 to the transmembrane AIMs in Rtn2 as the ER membranes become increasingly curved (*Figure 7C*).

TEX264 has been shown to concentrate in three-way ER junctions prior to its incorporation into autophagosomes (*An et al., 2019*), suggesting that phagophore loading during ER-phagy happens at specific subdomains of the ER. Maize Rtn2 also showed a preferential localization within three-way tubular ER junctions, but it did not change its localization upon induction of ER stress by DTT (*Figure 4A*). Instead, Rtn2 binds more Atg8 under ER stress, a property that has not been reported for any other known ER-phagy receptor. The increased binding of Rtn2 to Atg8a occurred within 3 hr of DTT treatment and required the C-terminal AIM of Rtn2. How exactly ER stress translates into increased binding of Rtn2 to Atg8 is presently unknown, but it does not seem to involve either the lipidation of Atg8 or the cysteine residues of Rtn2 located close to the lumenal face of the ER membrane (*Figure 2—figure supplement 2*; *Figure 4—figure supplement 1*).

Recently, calnexin was shown to act as a co-receptor with FAM134B to mediate the degradation of procollagen through ER-phagy, with calnexin binding misfolded procollagen polypeptides through its ER luminal domain and FAM134B through its transmembrane domain. FAM134B then binds Atg8/LC3 to sequester ER domains containing FAM134B, calnexin, and misfolded procollagen into autophagosomes for degradation (*Forrester et al., 2019*). However, the binding between FAM134B and calnexin was found to be stable and did not change with increasing concentrations of misfolded procollagen. We found here that maize Rtn2 binds calnexin and two additional proteins involved in protein folding, Bip and Pdi. As the three proteins became more abundant during DTT treatment, they also increased their association with Rtn2. A possible scenario is that the increasing number of calnexin, Bip, and/or Pdi molecules associated with Rtn2 under ER stress triggers either a conformational change or a post-translational modification within the C-terminal domain of Rtn2 to increase the accessibility and/or affinity of its AIM1 for Atg8 (*Figure 7C*).

We found that over-expression of Rtn2 in protoplasts altered ER morphology and reduced diffusion of an ER lumen marker, consistent with the formation of very narrow tubular constrictions like those generated by the over-expression of *Arabidopsis* RTNLB13 (*Tolley et al., 2008*). However, unlike mammalian FAM134B and yeast Atg40 (*Khaminets et al., 2015*), the overexpression of plant Rtn proteins does not seem to mediate ER fragmentation, which is needed to deliver ER domains into autophagosomes. The mechanism by which ER domains are fragmented during autophagy is still unclear, but a recent study showed that human atlastin 2 (ATL2), a GTPase that mediates homotypic fusion of the ER, is required for FAM134B-mediated ER-phagy and ER fragmentation (*Liang et al., 2018*). The *Arabidopsis* genome codes for three atlastin-like proteins named ROOT HAIR DEFECTIVE (RHD) 3, RHD3-L1, and RHD3-L2 involved in root development (*Chen et al., 2011*). Whether these are required for ER-phagy is currently unknown.

## Reticulons help regulate ER stress

The up-regulation of ER stress-responsive chaperones in aleurone cells of maize *rtn2* mutants suggests that Rtn2 helps repress ER stress in these cells. This response was unexpected considering that, at least in human kidney cells, RTN proteins are needed to trigger ER stress (*Fan et al., 2017*; *Xiao et al., 2016*; *Fan et al., 2015*). For example, RTN1A over-expression is sufficient to induce ER stress and apoptosis in renal cells, whereas reduced expression of RTN1A attenuates ER stress induced by tunicamycin, albumin, and hyperglycemia (*Xiao et al., 2016*; *Fan et al., 2015*). RTN1A promotes ER stress through its interactions with the ER stress sensor PERK. Both the N- and C-terminal domains of RTN1A interact with PERK and are needed for ER stress promotion (*Fan et al., 2015*). However, the positive role of RTN1A in ER stress might not extend to the rest of the RTN family. RTN1A is one of the three isoforms of mammalian RTN1 and contains an N-terminal domain of over 550 amino acids. RNT1C, another isoform of RTN1, has short N- and C-terminal domains similar to maize Rtn1 and Rtn2, and does not participate in ER-stress induction in kidney cells (*Fan et al., 2015*), suggesting that the long N-terminal extension of RTN1A is critical for engaging PERK1 and inducing ER stress. The specialized role of RTN proteins with long N-terminal extensions has also been documented in plants. For example, *Arabidopsis* RTNLB20 does not modify ER morphology when over-expressed, but its N-terminal extension is involved in sterol synthesis (*Kriechbaumer et al., 2018*). Whether the other plant Rtn proteins with long N-terminal extensions (*e.g.* maize Rtn22 and Rtn23) play similar roles to RTN1A in promoting ER stress remains to be determined. However, given that PERK-like proteins have not been identified in plants (*Ruberti et al., 2015*), it remains possible that plant Rtn proteins do not participate in ER stress induction.

It is notable that the up-regulation of ER stress genes such as *Bip2*, and the consequent increase in autophagic flux in *rtn2* mutants are more pronounced in aleurone cells than in the starchy endosperm. The starchy endosperm can experience ER stress, and, in fact, the plant UPR was first described in maize endosperm cells expressing mutant storage proteins that cause abnormal protein body assembly (*Marocco et al., 1991*; *Fontes et al., 1991*; *Boston et al., 1991*). Therefore, the stronger up-regulation of the ER stress responsive gene *Bip2* in *rtn2* aleurone cells might be connected to a more critical requirement for Rtn1/2 during ER homeostasis in aleurone versus starchy endosperm cells, as aleurone cells actively export ER protein accretions (*Reyes et al., 2011*) and remodel their ER to accommodate lipid droplets. Although we cannot determine at this time whether ER-stress in *rtn2* mutant aleurone cells is triggered by deficient ER-phagy or by an autophagy-independent Rtn2 function, it is remarkable that *Bip2* up-regulation in aleurone cells followed a very similar pattern in *rtn2*, *atg12*, and *rtn2 atg12* double mutants (*Figure 7B*), suggesting that specific requirements for efficient ER-phagy in aleurone cells could indeed be the common ER stress trigger in both *rtn2* and *atg12* mutants.

## Materials and methods

A Key Resource Table can be found in *Supplementary file 1*.

### Plant material and growth conditions

*Zea mays* lines W22, Mu-08559 (*rtn1-1*), Mu-02254 (*rtn2-1*), Mu-03723 (*rtn2-2*), and Mu-00381 (*rtn2-3*) were obtained from the Maize Genetics Cooperation Stock Center. Plants expressing YFP-Atg8a (*Li et al., 2015*) were used as pollen donors to introgress this fluorescent marker into the different mutant backgrounds. The genotypes of the various mutant and reporter lines were confirmed by PCR of genomic DNA. Primers for the genotyping of the *UniformMu* lines, RT-PCR, qRT-PCR, and cloning are listed in *Supplementary file 1*.

Mutant lines were backcrossed between 2 and 6 times to W22 and the final F1 plants were self-crossed to obtain the homozygous mutants. All lines together with their corresponding controls were grown under similar conditions in either a field at West Madison Agricultural Research Station, or in greenhouses under a 16 hr light/8 hr darkness photoperiod, with supplemental lighting provided at an intensity of 330 µmol $m^{-2}$ $s^{-1}$, and average temperatures of 28°C during the light period and 21°C during the dark period.

### Reverse transcriptase (RT)-PCR of *Rtn1* and *Rtn2* transcripts

Developing W22, *rtn1,* and *rtn2* seeds were collected from ears at approximately 20 days after pollination and immediately stored at −80°C. Aleurone layers were excised by mechanical peeling from defrosted kernels, and total RNA extraction was performed as previously described *Reyes et al. (2011)*. Between 2 and 4 μg of total RNA were reverse transcribed to the first strand of the cDNA, using random primers and a High Capacity cDNA Reverse Transcription Kit (Applied Biosystems by Thermos Fisher Scientific) in a 20 μl reaction volume. For PCR, 1 to 2 μl of the synthesized cDNA was used as template in a 20 μl volume reaction. The resulting PCR products of the *Rtn* cDNAs were then sequenced to characterize mutations.

### Quantitative real time (qRT)-PCR

Fresh endosperm samples were collected from developing seeds 20 days after pollination. Four independent samples of approximately 30 seeds each (~150 mg of tissue) were used for either the control or DTT treatments. The pericarp and embryo were removed from each seed, and the top 2/3 part of the remaining endosperm was collected and placed in either 10 ml of ½ MS medium (Murashige and Skoog Basal Salt Mixture [Sigma-Aldrich], pH 5.7), or in 10 ml ½ MS medium containing 2 mM DTT, inside a 50 ml falcon tube. The tubes were then laid horizontally in the dark at room temperature for 3 hr. The aleurone cell layer was excised by mechanical peeling under a dissecting microscope immediately after treatment and was rapidly frozen on dry ice. The collected aleurone cell layers were then stored at −80°C until used for qRT-PCR analysis.

For the quantification of *Bip2* (ER stress marker) and EF1α (reference gene) transcript levels, aleurone peels and starchy endosperm samples were dissected from frozen seeds of homozygous W22, *atg2-2*, *rtn2-2*, and *rtn2-2 atg12-2* ears at 20–21 days after pollination. Three or four independent samples of 30 seeds each were assayed for each genotype. Total RNA was extracted as previously described *Reyes et al. (2011)*, residual genomic DNA was removed using TURBO DNA-*free* Kit (Invitrogen), and reverse transcription was performed as explained above. Amplification and detection of target transcripts were performed using the MAXIMA SYBR Green/ROX qPCR Master Mix (Thermo Fisher Scientific). The qRT-PCR reactions were conducted in a final volume of 20 μl with a Stratagene 512 MX3000P qPCR system that monitored double strand DNA synthesis. All reactions were performed with a 58°C annealing temperature and 300 μM of each primer. Results were analyzed with LinRegPCR (version 2013.0; http://www.hartfaalcentrum.nl/index.php?main=files&sub=LinRegPCR) using at least three biological replicates that were each measured in triplicate. The relative expression levels of each gene were calculated by the comparative threshold cycle (Ct) method (*Schmittgen and Livak, 2008*) using the *EF1α* gene as reference. The ΔΔCt of each DTT-treated aleurone sample was obtained by calculating the difference of ΔCt (DTT-treated) minus ΔCt (corresponding control). The ΔΔCt value (*rtn2-2*, *atg12-2*, or *rtn2-2 atg12-2*) of each biological replicate per target transcript was obtained by calculating the average of the ΔCt (mutant) minus the ΔCt of each biological replicate for wild-type W22, respectively.

### DNA cloning, constructs, and plasmids

For the soluble Y2H assay, the cDNAs encoding maize *Atg8a* (Zm00001d006474_T001), or the N-terminal (amino acids 1 to 80), cytoplasmic loop (amino acids 124 to 173), and C-terminal (amino acids 206 to 263) regions of Rtn2 were first recombined into pDONR221 *via* the Gateway BP clonase II reaction, and then further recombined in-frame with coding regions of either the GAL4 activation domain or GAL4 DNA-binding domain in the pDEST22 or pDEST32 vectors, respectively, *via* the Gateway LR clonase II reaction.

For transforming protoplasts, all cDNAs sequences were cloned into the pRTL2-mCherry vector (*Arabidopsis* Biological Resource Center, stock number CD3-1062). The cDNAs of *Rtn2* and *Rtn1* were fused to *eGFP* by overlapping PCR and cloned into pRTL2 between the *Sma* I and *Xba* I restriction enzyme sites after removal of the *mCherry* sequence. Other constructions encoding Rtn2 fused to mCherry at the C terminus, wild-type and mutated Rtn2 with N-terminal HA tags, and eGFP-Atg8a (both wild type and mutant forms), were also cloned onto pRTL2.

## Yeast two-hybrid (Y2H) assays

Split-ubiquitin Y2H assays were performed using the DUAL membrane pairwise kit (Dualsystems, Biotech; catalog number P01501). The full coding sequences of wild type of *Rtn1*, *Rtn2*, and *Atg8a*, as well as *Rtn1* and *Rtn2* harbouring mutated AIM domains were amplified by or overlapping PCR, using Phusion High-Fidelity DNA Polymerase (Thermo Fisher Scientific) and cloned into pBT3-N, pBT3-C (*LEU2* auxotrophy, kanamycin resistance), pPR3-N or pPR3-C (*TRP1* auxotrophy, ampicillin resistance), using the *Sfi* I restriction site. Resulting combination of plasmids (including empty vectors as negative controls) were then co-transformed into yeast strain NMY51. The number of transformed colonies detected on synthetic dropout medium lacking leucine, tryptophan, histidine, and adenine (*i.e.*, selecting for the interaction) was divided by the number of transformed colonies detected on synthetic dropout medium lacking leucine and tryptophan only (*i.e.*, selecting for co-transformation) to calculate the percentage of colonies showing a positive interaction. Between 200 and 300 colonies were counted in each case.

Interactions between soluble peptides derived from Rtn2 and maize Atg8a were assessed using the ProQuest Y2H system (Thermo Fisher Scientific). Pairwise combinations of putative interacting partners in pDEST22 and pDEST32 (or the empty vectors as controls) were co-transformed into yeast strain MaV203. Cells transformed with both plasmids were selected by growth during 2 days at 30°C on synthetic dropout medium lacking leucine and tryptophan. Positive interactions were identified by growing transformed cells for 2 days at 30°C on synthetic dropout medium lacking leucine, tryptophan and histidine, and containing 25 mM 3-amino-1,2,4-triazole (3-AT). To confirm the interactions, single colonies were diluted in sterile water to an $OD_{600}$ of 0.1, and 5 mL was spotted onto both types of selective medium and grown for 2 days at 30°C.

## Transformation of maize and *Arabidopsis* protoplasts

Maize (*Zea mays*, inbred line B73 crossed to Mo17) F1 seeds were first germinated on filter paper, and then transferred into plant growth medium (Sungrow Horticulture Propagation Mix) and grown in the dark for 10 to 14 days at room temperature. At this stage, the seedlings usually had two leaves, both of which were used for protoplast isolation. Approximately 12 leaves from the etiolated maize seedlings were removed and cut into small pieces using a sharp razor blade without squeezing the leaves. The cut leaves were cultured in enzyme solution (3% [w/v] cellulose [Dot Scientific], 0.7% [w/v] macerozyme [Dot Scientific], 0.59M mannitol, 10 mM KCl, 10 mM MES [pH 5.7], 1 mM $CaCl_2$, 0.02% [w/v] BSA) for 3 hr in the dark at room temperature, with constant agitation (60 rpm). Released protoplasts were then filtered through a 35 µm mesh and pelleted by centrifugation at 150 x *g* for 5 min at 4°C in a round-bottomed tubes. Protoplasts were kept on ice after centrifugation. The enzyme solution was removed and the protoplasts were washed two times with 10 ml culture buffer (0.59 M mannitol, 10 mM KCl, 10 mM MES [pH 5.7], 1 mM $CaCl_2$, 0.02% [w/v] BSA). The protoplasts were then re-suspended in 2 to 3 ml culture buffer. For transformation, 30 to 40 µg of plasmid DNA, approximately 220 µl of culture buffer, and 250 µl of protoplasts (approximately $2 \times 10^5$ protoplasts in total) were added to a 4 mm electroporation cuvette, with a final volume of 500 µl. Electroporation was performed at 200 V and 200 µF (0.5-1kV/cm) for 5 msec. Protoplasts were left on ice for 10–15 min, and 500 µl culture buffer was added to the cuvette before transferring protoplasts into a 6-well culture plate containing 1 ml of culture buffer per well. Protoplasts were then cultured at room temperature in the dark for 18–22 hr before imaging.

Isolation of *Arabidopsis* protoplasts (ecotype Col-0) was performed as previously described *Wu et al. (2009)*. Between 4 and 14 µg of pRTL2 vectors and 20–25 µg of *pDOE8-MAS:CFP-KDEL* vector (*Gookin and Assmann, 2014*) were used for each transformation. Protoplasts for confocal microscopy imaging were incubated at 23°C in darkness for 12 to 14 hr before imaging. To visualize autophagic bodies under ER stress conditions, protoplasts were cultured in the presence of 2 mM DTT (dissolved in $H_2O$) with or without 1 µM ConA (dissolved in DMSO) before imaging. As controls, protoplasts were also treated individually with 2 mM DTT, DMSO, or 1 µM ConA.

Protoplasts for co-immunoprecipitation were incubated at 23°C in darkness for 12 to 14 hr, subjected to treatment with 2 mM DTT or 5 µg/mL tunicamycin (dissolved in DMSO) for 3 hr, and collected by centrifugation for 3 min at 100 x *g* in a bench-top centrifuge. Control protoplasts for tunicamycin treatment were treated with equal concentration of DMSO for 3 hr and collected as described before. After removing the buffer, protoplasts were re-suspended in 50 µl of protein

extraction buffer (20 mM Tris-HCl (pH 7.5), 100 mM NaCl, 20 mM ethylenediaminetetra-acetic acid (EDTA), 0.5% (v/v) Triton X-100, 5 mM NaF, 1 mM $Na_3VO_4$, 1 mM DTT, 1 mM phenylmethylsulfonyl fluoride (PMSF), 1X plant protease inhibitor cocktail), immediately frozen in liquid nitrogen, and stored at −80˚C until protein extraction.

For testing Rtn2 protein stability, 12 µg of pRTL2 vectors expressing HA-Rtn2 and HA-Rtn2 (mAIM1,2,5,6) were used for transforming Arabidopsis protoplasts. We incubated the transformed protoplasts at 23˚C in darkness for 12 hr, treated them with 200 µg/mL CHX, and collect samples at 0, 30, 60, and 180 min. Control samples (no CHX) were collected at 180 min after the onset of the CHX treatment. Protein were extracted as described above.

## Protein extraction and immunoblot analysis

Frozen maize aleurone and starchy endosperm samples were homogenized in 10 volumes of protein extraction buffer (20 mM Tris-HCl (pH 7.5), 100 mM NaCl, 20 mM ethylenediaminetetra-acetic acid (EDTA), 0.5% (v/v) Triton X-100, 5 mM NaF, 1 mM $Na_3VO_4$, 1 mM DTT, 1 mM phenylmethylsulfonyl fluoride (PMSF), 1X plant protease inhibitor cocktail (Sigma-Aldrich)), and clarified by centrifugation at 16,000 x $g$ for 5 min at 4˚C. The supernatant was then mixed with 0.25 volumes of 5X SDS-PAGE sample buffer (200 mM Tris-HCl (pH 6.8), 25% (v/v) glycerol, 10% (w/v) SDS, 10% (v/v) β-mercaptoethanol, 0.1% (w/v) bromophenol blue). Samples were heated at 95˚C for 5 min and then subjected to SDS-PAGE analysis, with 10 to 20 µl of each sample being run on linear gels containing 10, 12 or 14% acrylamide, as appropriate, followed by immunobloting with the indicated antibodies.

Frozen protoplasts were homogenized by addition of 400 µl homogenization buffer (150 mM Tris-HCl (pH 7.5), 150 mM NaCl, 1.5% (v/v) Triton X-100, 2 mM PMSF, 1X plant protease inhibitor cocktail) followed by vigorous mixing with a vortex, and clarified twice by centrifugation at 16,000 x $g$ for 5 min at 4˚C. Equal volumes of supernatant (300 µl) were then incubated for 2 hr at 4˚C with 50 µl EZview red anti-HA affinity gel (100 µl of a 50% slurry; Sigma-Aldrich) or 25 µL GFP-Trap agarose beads (50 µL of a 50% slurry, Chromotek), pre-equilibrated in homogenization buffer. Beads were then collected by centrifugation at 6000 x $g$ for 1 min at 4˚C, washed five times with ice-cold homogenization buffer, and bound proteins were eluted in 100 µl 2X SDS-PAGE sample buffer (80 mM Tris-HCl (pH 6.8), 10% (v/v) glycerol, 4% (w/v) SDS, 4% (v/v) β-mercaptoethanol, 0.04% (w/v) bromophenol blue) by heating at 95˚C for 5 min. Samples were then analyzed by SDS-PAGE as above, followed by immunoblotting with the indicated antibodies.

For immunoblot analyses, proteins separated by SDS-PAGE were electrophoretically transferred onto Immobilon-P polyvinylidene difluoride (PVDF) membranes (Millipore) for 16 hr at 80 mA, and the membrane was blocked for at least 60 min with a 10% (w/v) non-fat dry milk solution in 1X PBS (137 mM NaCl, 2.7 mM KCl, 10 mM $Na_2HPO_4$, 1.8 mM $KH_2PO_4$), which was first filtered through two layers of Miracloth. All incubations were performed at room temperature. The membrane was incubated with primary antibody solution (in 1% (w/v) non-fat dry milk solution in PBS) for 60 min, before being washed once with PBS, once with PBST (PBS containing 0.1% (v/v) Triton X-100), and once with PBS for 10 min each. The membrane was re-blocked with 10% (w/v) non-fat dry milk solution in PBS for 30 min, incubated for 60 min with secondary antibody solution (in 1% (w/v) non-fat dry milk solution in PBS), and then washed again as above.

Primary antibodies used were anti-Aca2 (1:1,000) (*Harper et al., 1998*; *Reyes et al., 2011*), anti-Atg8a (1:500) (*Thompson et al., 2005*), anti-Bip (1:3,000) (*Holding et al., 2007*), anti-calnexin/calreticulin (1:5,000) (*Pagny et al., 2000*), anti-GFP (1:5,000; Sigma-Aldrich, product number 11814460001), anti-HA (1:5,000; Sigma-Aldrich, product number H6908) anti-H3 (1:3000 for protoplast samples and 1:10,000 for aleurone and endosperm samples; AbCam, product number AB1791), anti-Nbr1 (1:1,000) (*McLoughlin et al., 2018*), anti-Pdi (1:5,000) (*Li and Larkins, 1996*), anti-Rpt4 (1:3,000) (*Marshall et al., 2015*), and anti-Vdac (1:1,000) (*Subbaiah et al., 2006*). Secondary antibodies used were goat anti-mouse HRP conjugate (1:5,000–10,000; SeraCare, product number 074–1806), goat anti-rabbit HRP conjugate (1:5,000–10,000; SeraCare, product number 074–1506), or rabbit anti-chicken IgY HRP conjugate (1:3,000) (*Rancour et al., 2004*).

Blots of three independent biological replicates were developed using the SuperSignal West Pico Plus Chemiluminescent Substrate or the SuperSignal West Femto Maximum Sensitivity Substrate (both from Thermo Fisher Scientific). Densitometric quantifications were performed using TotalLab software (Non-linear Dynamics), using three different exposures of each blot to ensure that the exposure level was within the linear range of the film.

## Confocal microscopy on maize aleurone peels and protoplasts

All fluorescence microscopic images were captured on either a 710 or 780 Zeiss laser scanning confocal microscope. Seeds expressing YFP-Atg8a were collected at 19–21 days after pollination and the aleurone cell layer was dissected by mechanical peeling. Aleurone cells were imaged using a 40x objective (numerical aperture 1.1). YFP was excited with a 514 nm excitation laser line and detected using a 519–544 nm band-pass filter.

Maize and *Arabidopsis* leaf protoplasts were loaded onto an 18 Well Flat μ-Slide (Ibidi, Cat.No. 81826) and imaged using a 63x water immersion objective (numerical aperture 1.46). GFP was excited with a 488 nm laser line and detected using a 490–526 nm band-pass filter, mCherry was excited with a 561 nm laser line and detected using a 576–621 nm band-pass filter, and CFP was excited with a 405 nm laser line and detected using a 454–536 nm band-pass filter.

The multitrack mode was used for sequentially imaging of GFP and mCherry, CFP and mCherry. The emission spectra of GFP, CFP, YFP, and mCherry were confirmed by spectral scans. Colocalization between Rtn2-mCherry and CFP-KDEL was quantified using Pearson's colocalization coefficients in the ImageJ software (*Abràmoff et al., 2004*).

## Transmission electron microscopy

Small pieces of endosperm containing both aleurone and starchy endosperm cells were excised from W22, *rtn1-1, rtn2-1, rtn2-2, rtn2-3, atg12-2,* and *rtn2-3 atg12-2* mutant seeds at 19–20 days after pollination and high-pressure frozen in a Baltec 010 HPM high pressure freezer. Frozen samples were freeze-substituted at −80°C in 2% OsO₄ overnight. Samples were then embedded in Epon resin, sectioned, and stained with 2% uranyl acetate in 70% methanol and lead citrate (2.6% lead nitrate and 3.5% sodium citrate [pH 12]).

## Phylogenetic analysis and expression pattern of plant reticulons

Amino acids sequences from RHD-containing plant proteins were obtained from the *Arabidopsis* Information Resource (http://www.arabidopsis.org/), Phytozome (http://www.phytozome.net/), and the Maize Genetics and Genomics Database (https://www.maizegdb.org). Protein sequence alignments were performed using ClustalW (http://www.ebi.ac.uk/Tools/msa/clustalw2/). The evolutionary history was inferred by using the Maximum Likelihood method based on the JTT matrix-based model (*Jones et al., 1992*). The bootstrap consensus tree inferred from 100 replicates was taken to represent the evolutionary history of the taxa analyzed. Branches corresponding to partitions reproduced in less than 50% of bootstrap replicates were collapsed. The percentage of replicate trees in which the associated taxa clustered together in the bootstrap test (100 replicates) are shown next to the branches (*Felsenstein, 1985*). Initial tree(s) for the heuristic search were obtained automatically by applying Neighbor-Join and BioNJ algorithms to a matrix of pairwise distances estimated using a JTT model, and then selecting the topology with a superior log likelihood value. The analysis involved 62 amino acid sequences with a total of 980 positions in the final dataset. Evolutionary analyses were conducted in MEGA7 (*Kumar et al., 2016*).

Developmental and tissue expression patterns were obtained from for rice (http://bar.utoronto.ca/efprice/cgi-bin/efpWeb.cgi) and maize B73 inbred lines (http://bar.utoronto.ca/efp_maize/cgi-bin/efpWeb.cgi?dataSource=Sekhon_et_al_Atlas) using public databases.

## Acknowledgements

We would like to thank Edward Wilkinson and Jaydin Grenier for their assistance during this project, Natalia de Leon (University of Wisconsin) and the staff at the West Madison Agricultural Research Station and at the Wisconsin Crop Innovation Center for their support growing and processing maize material, Rebecca Boston (North Carolina State University) for providing the anti-calnexin antibodies, and Donald McCarty and Karen Koch (University of Florida) for developing the *UniformMu* lines. This work was supported by National Science Foundation grants IOS-1339325 and IOS-1840687 to MSO and RDV, the United States Department of Agriculture; National Institute of Food and Agriculture Hatch Act Formula Fund WIS01791 to MSO, and funds from the University of Wisconsin; Department of Botany to XD.

## Additional information

### Funding

| Funder | Grant reference number | Author |
|---|---|---|
| National Science Foundation | IOS-1840687 | Marisa S Otegui |
| Agriculture Hatch Act Formula Fund | WIS01791 | Marisa S Otegui |
| National Science Foundation | IOS-133932 | Richard David Vierstra |
| U.S. Department of Agriculture | | Marisa S Otegui |
| National Institute of Food and Agriculture | Hatch Act Formula Fund WIS0179 | Marisa S Otegui |
| University of Wisconsin-Madison | | Xinxin Ding |

The funders had no role in study design, data collection and interpretation, or the decision to submit the work for publication.

### Author contributions

Xiaoguo Zhang, Conceptualization, Data curation, Formal analysis, Supervision, Validation, Investigation, Methodology; Xinxin Ding, Conceptualization, Data curation, Formal analysis, Validation, Investigation, Visualization, Methodology; Richard Scott Marshall, Conceptualization, Formal analysis, Validation, Investigation, Visualization, Methodology; Julio Paez-Valencia, Formal analysis, Validation, Investigation, Methodology; Patrick Lacey, Validation, Investigation; Richard David Vierstra, Conceptualization, Supervision, Funding acquisition, Investigation, Methodology; Marisa S Otegui, Conceptualization, Formal analysis, Supervision, Funding acquisition, Validation, Investigation, Visualization, Methodology, Project administration

### Author ORCIDs

Richard Scott Marshall (iD) https://orcid.org/0000-0002-6844-1078
Richard David Vierstra (iD) https://orcid.org/0000-0003-0210-3516
Marisa S Otegui (iD) https://orcid.org/0000-0003-4699-6950

### Decision letter and Author response

Decision letter https://doi.org/10.7554/eLife.51918.sa1
Author response https://doi.org/10.7554/eLife.51918.sa2

## Additional files

### Supplementary files

- Supplementary file 1. Key resources table.

- Transparent reporting form

### Data availability

All data generated or analysed during this study are included in the manuscript and supporting files.

The following previously published dataset was used:

| Author(s) | Year | Dataset title | Dataset URL | Database and Identifier |
|---|---|---|---|---|
| Buell CR | 2013 | Zea mays subsp. mays transcriptome or gene expression, B73 18DAP Embryo. | https://trace.ncbi.nlm.nih.gov/Traces/sra/?study=SRP014652 | NCBI Sequence Read Archive, SRP014652 |

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
