## [Decision Letter]

**Acceptance summary:**

In the past few years, autophagy, and in particular, reticulophagy (ER removal and degradation) has emerged as critical mechanism for maintaining homeostasis in yeast, plant and mammalian cells. In this work, the authors identified putative homologues of the mammalian RTN3L protein in maize, and used an impressive variety of assays in vitro and in whole plants to probe the cellular role of two plant reticulon proteins (Rtn1 and Rtn2) in ER homeostasis and reticulophagy. The functions and behaviors of the plant proteins were mostly shared with RTN3L in mammals, including interactions with Atg8, however they also identified some unique structural and regulatory features of the interaction. Their analysis of maize mutants revealed that Rtn1 and Rtn2 are required to ameliorate ER stress at a critical point in development. In specialized aleurone cells that coordinate the mobilization of nutrients to the embryo during germination loss of Rtn1/2 alters bulk autophagy. This work, therefore, is exciting from both a basic science and potentially an agronomic viewpoint.

**Decision letter after peer review:**

Thank you for submitting your article "Reticulon proteins modulate autophagy of the endoplasmic reticulum in maize endosperm" for consideration by *eLife*. Your article has been reviewed by three peer reviewers, one of whom is a member of our Board of Reviewing Editors, and the evaluation has been overseen by a Reviewing Editor and Christian Hardtke as the Senior Editor. The reviewers have opted to remain anonymous.

The reviewers have discussed the reviews with one another and the Reviewing Editor has drafted this decision to help you prepare a revised submission.

Summary:

In the past few years, autophagy, and in particular, reticulophagy (ER removal and degration) has emerged as critical mechanism for maintaining homeostasis in yeast, plant and mammalian cells. Several reticulophagy receptors in mammals and their mechanisms of action are an area of active interest. After identifying putative homologues of the mammalian RTN3L protein in Maize, the authors use a number of biochemical and protoplast assays to probe the cellular role of two plant reticulon proteins in ER homeostasis and reticulophagy. Analysis of Maize mutants suggests a role in preventing ER stress in the seed endosperm.

Overall, reviewers found the topic important from a basic science and potentially an agronomic viewpoint. The findings here were mostly similar to those from studies of RTN3L in mammals, which isn't to say that the findings here aren't interesting or important. Two expert reviewers did a very thorough analysis of the text and figures and recommend a number of critical control experiments to solidify the work.

Essential revisions:

1) The use of heterologous systems to test the Rtn1 and Rtn2 proteins was deemed acceptable, but necessitates additional controls. Details on such controls are described nicely in reviewer 2, major comment 1.

2) More evidence (by additional or improved IPs) of the interactions and stoichiometry of the complex is needed.

3) It needs to be clarified how AIM motifs buried in the intramembrane regions of the rtn1/2 proteins could be functional for binding.

4) The quality of the figures (resolution) in the main text and the supplemental must be improved. We recognize some of the compression during journal file production exacerbated this issue, but there were several concerns about blots in the supplemental and low resolution and/or low magnification images of vacuoles and Atg8a puncta.

5) The overall flow of the Introduction could be improved. Reviewer 3 has some suggestions for this; it is not necessary to format in precisely the way they suggestion, but please take note of the reasoning behind their suggestions. In addition, sticking to conventional nomenclature will make this work more easily integrated and evaluated within the field.

6) In your revision, please include a response to the experiments described in the major comments from the reviewers-if a control cannot or was not done, please explain why and also consider rewording conclusions if the experiment cannot be done.

Reviewer #2:

1) I think it is totally fine to use heterologous systems. However they may have some limitations that need to be considered. As the authors express maize RTNs in Arabidopsis and RTNs are well known to form higher order structures, do maize RTNs also interact with Arabidopsis RTNs? If they do, it would be difficult to interpret protoplast coIP assays and flux assays, as Arabidopsis RTNs are also likely to function as ER-phagy receptors. Consistently, the results in Figure 3K suggest, the AIM mutant is still recruited to the autophagosomes. Is this due to the interaction with the Arabidopsis RTNs?

2) Why did the authors focus on RTN1 and RTN2? Is there any other reason than higher expression in the seeds? In RTN3L, Dikic and colleagues clearly showed the extended region contained the multiple AIMs, explaining why this specific RTN was mediating reticulophagy. Is this the case of RTN1 and RTN2? Are the AIMs specific to these two isoforms? If yes, the manuscript would really benefit from using another RTN as a negative control.

3) As the authors also explain in the text, all the AIMs in the known ER-phagy receptors are in exposed regions, not buried in transmembrane domains. Although the authors tried to come up with an explanation like stretching exposing these AIMs, I think they still need to strengthen this part. I don't know how DTT treatments could stretch the membranes. As in their heterologous systems, they overexpress their proteins; another possible explanation could be misfolded proteins, rather than membrane stretching. Have the authors tested the stability of the AIM mutants? In Figure 2F, the protein levels in the input are not the same. Is this due to changes in protein folding and/or stability?

4) Figure 2B, where the authors showed the interaction with the ATG8A using HA pull downs, should be supported with reverse pull downs. This way, we will be able to know if ATG8 is interacting with RTN monomers or oligomers. Figure 2F would also benefit from reverse IP. These reverse IPs would also be very useful in the DTT treated coIP assays.

5) Also in Figure 2F, could the authors please check if the input and IP HA blots are switched? The input bands look stronger than the IPed ones and compared to Figure 2B, the oligomer bands are much weaker.

6) Figure 3K: the induction with DTT seems very minor without ConA. With ConA, colocalization is very difficult as particles inside of the vacuoles move so fast. The authors could perform time course experiments to test if extending the DTT treatment would increase colocalization.

7) The experiments where the authors tested if RTNs also need a coreceptor desperately need a negative control. BiPs, PDIs are well known to be sticky proteins that always interact with protein of interest. As their concentration will increase upon DTT treatment, the intensity of the bands in the IPs could also interact. The authors should use an ER-membrane protein that does not contain AIMs as a negative control in these experiments.

8) Also the western blots in the supplementary figures look highly modified. Different lanes have different backgrounds and in some cases different bands look identical, at least in the version that we could analyze (Figure 2—figure supplement 1: In the GFP input, lanes 4 and 5 seems identical. Figure 4—figure supplement 1 GFP input 7 and 8th lanes are identical). Could the authors please clarify this? I would strongly suggest that the authors add the full blots as a supplementary figure.

Reviewer #3:

1) The data presented in this study showed the identification of Rtn proteins involved in ER-phagy in maize. it was published the AtSEC62 ER-phagy receptor is the first case characterized in plants. I suggest including this example and then reorganize the Introduction because is too extended and not clear. Also the concept of reticulophagy receptors needs to be more clearly and broadly defined in the Introduction as this is the major focus of this study.

2) It is important to clarify How AIM motifs within intramembrane regions of the RHD domain in Rtns proteins could be functional. How this motif is available to interact with Atg8a? Are the constructs localized in the ER membranes? Even though it was mentioned in the text and tested by co-immunoprecipitation experiments is not enough evidence to a successful insertion into the ER membranes. This is an important point to consider and clarify and also how these proteins are folded if they were mutated for alanine residues and especially in these hydrophobic regions.

3) The authors mainly tested DTT as an unfolding protein response trigger. Generally, in this kind of studies, will be important to include other stressors as thapsigargin due to the important role of ER during calcium homeostasis that will be nicely correlated with the increased interaction with the chaperones under stress. These chaperones are mainly calcium sensitive. For other hand, could be important and interesting to include the stress abiotic as a model for ER stress induction. Cereal crops are usually exposed to abiotic stress that cannot be avoided. Hence interesting mechanisms of tolerance and adaptability could be evidenced when ER-phagy will be involved. Additionally, the identification of the Rtn proteins was in cereal cells, could be possible that the function of these proteins change between the different tissues? i.e. mesophyll cells protoplasts versus aleurone or endosperm cells.

4) The authors are advised to follow the nomenclature of published receptors that mediate ER remodelling. The process of reticulophagy was described some 10 years ago as a response of ER to the stress. The work of Walter's lab coined the nomenclature ER-phagy by describing the details of genetic and molecular characterization of this process. Since 2015 there are 8 distinct receptors that can mediate ER-phagy published including those from the yeast, plants and mammalian systems. In each manuscript the nomenclature was that these receptors were called ER-phagy receptors. Thus stating now that RTN1 and 3 from maize are reticulophagy receptors only contributes to the confusion of the readers who are not directly from this field.

General technical comments

1) To improve the manuscript the authors should reorganize the Introduction and the Figure 1 including some of the recent work published and the phylogeny analyses.

2) As was mentioned before will be important to include some tools to measure the autophagy flux and specific could use the same mCherry-GFP Rtn proteins as a reporter for ER-phagy.

3) There are some figures with low quality and with visible resolution problems in its pictures. Also, should be more clear if the authors can prepare some magnifications according to show nicely the morphology of vacuoles and the puncta formation positive to Atg8a.

4) Subsection “Rtn1 and Rtn2 regulate macroautophagy” it is not clear the conclusion. The authors should clarify this point where the deletion of rtn2-2 in a background defective in atg12-2 affects the vacuole contains. However in the text is indicated that the loss of Rtn2 increases Atg8 macro-autophagy. These data are not correlated with the western blots of the YFP-Atg8a for example.

[Editors' note: further revisions were suggested prior to acceptance, as described below.]

Thank you for resubmitting your work entitled "Reticulon proteins modulate autophagy of the endoplasmic reticulum in maize endosperm" for further consideration by *eLife*. Your revised article has been evaluated by Christian Hardtke (Senior Editor), Dominique Bergmann (Reviewing editor) and two reviewers.

The manuscript has been significantly improved, and we are moving toward acceptance, but there are some remaining issues that need to be addressed in the text before acceptance, as outlined below:

1) One statement that is still confusing and not clearly answered is how can it be that the interaction between ATG8 and RTN1/2 is mediated via the AIM that is localized into the transmembrane region of RTNs? Reviewers felt this was still not addressed in the revision.

2) It was suggested that in Figure 1, because Maize Rtn1 and Rtn2 are reticulon proteins. The diagrams comparing the position of the AIM domains at least with the mammalian RTN3L, yeast Atg40 and Maize Rtn1/2 could be added in the main figure.

3) Reviewers felt that it was not truly demonstrated that an increase in the Atg8 protein levels is a consequence of increased autophagy directly rather than that ER-stress triggers the upregulation of Atg8 and autophagy core machinery genes (Figure 5-6). This caveat should be made.

---

## [Author Response]

Essential revisions:1) The use of heterologous systems to test the Rtn1 and Rtn2 proteins was deemed acceptable, but necessitates additional controls. Details on such controls are described nicely in reviewer 2, major comment 1.

As explained in more detail below, we are aware that the Arabidopsis and maize Rtn proteins could have redundant functions and be able to interact with each other. This is not only a problem when using Arabidopsis protoplasts in a heterologous system but also when using maize cells. There are 23 Rtn proteins in maize and 21 in Arabidopsis; with the exception of maize Rtn1 and Rtn2, we currently do not know how many other Rtn proteins could act as ER-phagy receptors during ER stress. We still think that the results from the heterologous protoplast system are informative because we have been very careful in expressing comparable amounts of wild type and mutated Rtn2 proteins in all our experiments. We have done this by using equivalent amount of plasmid DNA and by controlling protein content by western blot. Therefore, even though the maize Rtn2 proteins with mutated AIMs may be recruited to autophagosomes at a low level through interactions with endogenous Arabidopsis Rtn proteins, the drastic decrease in their recruitment is informative of the functionality of these AIM domains. In addition, the fact that these proteins show reduced interaction with Atg8 in a yeast-two hybrid assay further support the notion that these domains are functional AIMs. We have made changes in the text to make the reader aware of this potential caveat in our protoplast assays (subsection “Rtn1 and Rtn2 as autophagic receptors for ER turnover during ER stress”).

2) More evidence (by additional or improved IPs) of the interactions and stoichiometry of the complex is needed.

We have now performed the reciprocal IP requested by the reviewer (Figure 2—figure supplement 2C of the revised manuscript). We co-expressed Rtn2-HA and GFP-Atg8a with or without DTT treatment and used anti-GFP antibodies for the IP. The new results confirm the interaction between Rtn2 and Atg8.

3) It needs to be clarified how AIM motifs buried in the intramembrane regions of the rtn1/2 proteins could be functional for binding.

Both reviewers expressed concerns about the stability and ER localization of the Rtn2 with alanine substitutions in the predicted transmembrane AIM domains. To address these concerns, we have included in the revised manuscript two new assays:

1) We tested the stability of the mutant and wild type Rtn2 proteins by treating protoplasts with cycloheximide (CHX) to inhibit protein synthesis and detecting protein content by western blotting over time. These results are now included in Figure 2H. Based on this assay, we found that HARtn2(mAIM1,2,5,6) seemed to be more stable than HA-Rtn2 after 180 min of CHX treatment. However, when the same CHX chase assay was done in Arabidopsis *atg7* mutant protoplasts, which are unable to perform macroautophagy, both HA-Rtn2 and HA-Rtn2(mAIM1,2,5,6) showed similar stability. These new results indicate that a) the mutations in the AIM domains located at transmembrane domains do not render Rtn2 unstable; b) the identified AIM domains are required for the autophagic degradation of Rtn2 during cycloheximide treatment.

2) We performed a colocalization analysis between the three Rtn2 proteins fused to mCherry with the ER lumenal marker CFP-ER (now included in Figure 2—figure supplement 1C of the revised manuscript). We found no differences in the Pearson’s colocalization coefficients for the three Rtn2-m-Cherry fusions co-expressed with similar amounts of CFP-KDEL. Based on the fact that wild type and AIM-mutated Rtn2 proteins a) show similar or even higher stability (Figure 2H); 2) show similar ER localization (Figure 2—figure supplement 1C); b) are equally able to interact with Rtn1 at the ER (Figure 2—figure supplement 1B), we assume that the AIM-mutated proteins are properly folded and inserted in the ER, indicating that the AIM domains at the transmembrane domains are indeed important for the interaction with Atg8 and ER-phagy.

4) The quality of the figures (resolution) in the main text and the supplemental must be improved. We recognize some of the compression during journal file production exacerbated this issue, but there were several concerns about blots in the supplemental and low resolution and/or low magnification images of vacuoles and Atg8a puncta.

This was an unintended consequence of embedding the figures in the text for the original submission. We are now submitting high-resolution versions of all figures. We have also provided the raw data for all panels and figures in this manuscript.

5) The overall flow of the Introduction could be improved. Reviewer 3 has some suggestions for this; it is not necessary to format in precisely the way they suggestion, but please take note of the reasoning behind their suggestions. In addition, sticking to conventional nomenclature will make this work more easily integrated and evaluated within the field.

We have reorganized the Introduction, added new citations, and replaced the word “reticulophagy” with “ER-phagy” in the revised manuscript, as requested by the reviewers.

6) In your revision, please include a response to the experiments described in the major comments from the reviewers-if a control cannot or was not done, please explain why and also consider rewording conclusions if the experiment cannot be done.

We have answered point by point all the suggestions and comments made by the two reviewers

Reviewer #2:1) I think it is totally fine to use heterologous systems. However they may have some limitations that need to be considered. As the authors express maize RTNs in Arabidopsis and RTNs are well known to form higher order structures, do maize RTNs also interact with Arabidopsis RTNs? If they do, it would be difficult to interpret protoplast coIP assays and flux assays, as Arabidopsis RTNs are also likely to function as ER-phagy receptors. Consistently, the results in Figure 3K suggest, the AIM mutant is still recruited to the autophagosomes. Is this due to the interaction with the Arabidopsis RTNs?

As the reviewer points out, most likely the Arabidopsis and maize Rtn proteins have redundant functions and are able to interact with each other. This is not only a potential problem when using Arabidopsis protoplasts in a heterologous system but also when using maize cells. There are 23 Rtn proteins in maize and 21 in Arabidopsis; with the exception of maize Rtn1 and Rtn2, we currently do not know how many other Rtn proteins could act as ER-phagy receptors during ER stress. We still think that the results from the heterologous protoplast system are informative because we have been very careful in expressing comparable amounts of wild type and mutated Rtn2 proteins. We have done this by using equivalent amount of plasmid DNA and by controlling protein content by western blot. Therefore, even though the Rtn2 proteins with mutated AIMs may be recruited to autophagosomes at a low level through interactions with endogenous Rtn proteins, the drastic decrease in their recruitment is informative of the functionality of these AIM domains. In addition, the fact that these proteins show reduced interaction with Atg8 in a yeast-two hybrid assay further support the notion that these domains are functional AIMs. We have made changes in the text to make the reader aware of this potential caveat in our protoplast assays (subsection “Rtn1 and Rtn2 as autophagic receptors for ER turnover during ER stress”).

2) Why did the authors focus on RTN1 and RTN2? Is there any other reason than higher expression in the seeds? In RTN3L, Dikic and colleagues clearly showed the extended region contained the multiple AIMs, explaining why this specific RTN was mediating reticulophagy. Is this the case of RTN1 and RTN2? Are the AIMs specific to these two isoforms? If yes, the manuscript would really benefit from using another RTN as a negative control.

We focused on Rtn1 and Rtn2 because of their relatively high expression in endosperm. Different from RTN3L, Rtn1 and Rtn2 do not have an extended N-terminal region (see first paragraph of Results). This is the case for the products of the three splicing versions of the *Rtn1* transcript and the predicted product of the only known transcript of *Rtn2*. Maize Rtn3 to Rtn18 have also short N- and C-terminal domains. All these 18 Rtn proteins plus the additional five Rtn proteins with extended N- or C-terminal domains in maize have predicted AIM domains, some on the same regions as Rtn1 and 2, some in other regions. However, the *in silico* prediction of AIM domains is highly unreliable and needs to be confirmed by Atg8-binding assays. Therefore, we did not have any reliable way to predict which Rtn protein could be used as a meaningful negative control.

3) As the authors also explain in the text, all the AIMs in the known ER-phagy receptors are in exposed regions, not buried in transmembrane domains. Although the authors tried to come up with an explanation like stretching exposing these AIMs, I think they still need to strengthen this part. I don't know how DTT treatments could stretch the membranes. As in their heterologous systems, they overexpress their proteins; another possible explanation could be misfolded proteins, rather than membrane stretching. Have the authors tested the stability of the AIM mutants? In Figure 2F, the protein levels in the input are not the same. Is this due to changes in protein folding and/or stability?

As explained paragraph three of subsection “Involvement of reticulons in ER-phagy”, the RHD domains of reticulon proteins can induce membrane curvature by creating membrane asymmetry through the stretching of the cytoplasmic leaflet of the ER (Bhaskara et al., 2019). This membrane deformation effect is enhanced by oligomerization mediated by interactions through the transmembrane domains. It is possible that once the RHD domains induce stretching on the cytoplasmic leaflet of the ER membrane, Atg8 gains increased access to the transmembrane AIMs in Rtn2 as the ER membranes become increasingly curved.

To answer the reviewer’s question, we have tested the stability of the mutant and wild type Rtn2 proteins by treating protoplasts with cycloheximide (CHX) to inhibit protein synthesis and chasing protein content over time. These results are now included in Figure 2H. Based on this assay, we found that HA-Rtn2(mAIM1,2,5,6) seemed to be more stable than HA-Rtn2 after 180 min of CHX treatment. However, when the same CHX chase assay was done in Arabidopsis *atg7* mutant protoplasts, which are unable to perform macroautophagy, both HA-Rtn2 and HARtn2(mAIM1,2,5,6) showed similar stability. These new results indicate that a) the mutations AIM domains located at transmembrane domains do not render Rtn2 unstable; b) the identified AIM domains are required for the autophagic degradation of Rtn2 during cycloheximide treatment. We also performed a colocalization analysis between the three Rtn2 proteins fused to mCherry with the ER lumenal marker CFP-ER (now included in Figure 2—figure supplement 1C of the revised manuscript). We found no differences in the Pearson’s colocalization coefficients for the three m-Cherry fusions co-expressed with similar amounts of CFP-KDEL. Based on the fact that wild type and AIM-mutated Rtn2 proteins a) show similar or even higher stability (Figure 2H); b) show similar ER localization (Figure 2—figure supplement 1C); c) are equally able to interact with Rtn1 at the ER (Figure 2—figure supplement 1B), we assume that the AIM-mutated proteins are properly folded and inserted in the ER, indicating that the AIM domains at the transmembrane domains are indeed important for the interaction with Atg8.

4) Figure 2B, where the authors showed the interaction with the ATG8A using HA pull downs, should be supported with reverse pull downs. This way, we will be able to know if ATG8 is interacting with RTN monomers or oligomers. Figure 2F would also benefit from reverse IP. These reverse IPs would also be very useful in the DTT treated coIP assays.

We have now performed the reciprocal IP requested by the reviewer (Figure 4—figure supplement 2 in the revised manuscript). We co-expressed Rtn2-HA and GFP-Atg8a with or without DTT treatment and used anti-GFP antibodies for the IP. The new results confirm the interaction between Rtn2 and Atg8 shown by the co-immunoprecipitations with anti-HA antibodies.

5) Also in Figure 2F, could the authors please check if the input and IP HA blots are switched? The input bands look stronger than the IPed ones and compared to Figure 2B, the oligomer bands are much weaker.

We have double-checked and there was no inadvertent switching of blots in this experiment. The reviewer raises an interesting point regarding the intensity and appearance of the oligomeric bands between different experiments (sometimes more apparent than others, sometimes more a single band and other times more of a smear); at this time, we do not know the reasons behind this.

6) Figure 3K: the induction with DTT seems very minor without ConA. With ConA, colocalization is very difficult as particles inside of the vacuoles move so fast. The authors could perform time course experiments to test if extending the DTT treatment would increase colocalization.

Figure 3K shows the percentage of protoplasts with autophagic bodies inside the vacuole that contain both mCherry (from Rtn2 proteins fused to mCherry) and GFP (from GFP-Atg8) signals. When we treat protoplasts with DTT but no ConA, we cannot detect autophagic bodies because they are quickly degraded inside the vacuole. Treatment with ConA stabilizes autophagic bodies by decreasing/blocking vacuolar degradation and therefore, it facilitates their visualization. With fast scanning settings, we had no problem capture the autophagic bodies inside the vacuoles. The main purpose of Figure 3K is to show that under similar ER stress conditions (DTT-treatment) and when vacuolar degradation is blocked (ConA treatment), there is a decrease in the percentage of protoplasts with GFP-Atg8-positive autophagic bodies that contained AIM-mutated Rtn2-mCherry proteins compared to those that contain wild type Rtn2 fused to mCherry.

7) The experiments where the authors tested if RTNs also need a coreceptor desperately need a negative control. BiPs, PDIs are well known to be sticky proteins that always interact with protein of interest. As their concentration will increase upon DTT treatment, the intensity of the bands in the IPs could also interact. The authors should use an ER-membrane protein that does not contain AIMs as a negative control in these experiments.

We have now repeated the co-immunoprecipitation of HA-Rtn2 in Arabidopsis protoplasts and detected the presence of two ER-resident proteins, the calcium channel ACA2 (ER membrane) and CFP-KDEL (ER lumen). Although both proteins (ACA2 and CFP-KDEL) were readily detected in the input fraction, we were unable to detect them in the co-immunoprecipitate (even after long exposures), suggesting that the association between HA-Rtn2 and the ER chaperones is indeed specific. This new co-immunoprecipitation is now shown in Figure 4—figure supplement 2 of the revised manuscript.

8) Also the western blots in the supplementary figures look highly modified. Different lanes have different backgrounds and in some cases different bands look identical, at least in the version that we could analyze (Figure 2—figure supplement 1: In the GFP input, lanes 4 and 5 seems identical. Figure 4—figure supplement 1 GFP input 7 and 8th lanes are identical). Could the authors please clarify this? I would strongly suggest that the authors add the full blots as a supplementary figure.

We have taken extreme care to provide the best possible quality western blots since they are critical for quantification. Close inspection of the high-resolution images will show that the bands indicated by the reviewer may be similar but not all identical. The original images containing the full western blots have been provided to the journal office for their independent scrutiny.

Reviewer #3:1) The data presented in this study showed the identification of Rtn proteins involved in ER-phagy in maize. It was published the AtSEC62 ER-phagy receptor is the first case characterized in plants. I suggest including this example and then reorganize the Introduction because is too extended and not clear. Also the concept of reticulophagy receptors needs to be more clearly and broadly defined in the Introduction as this is the major focus of this study.

We have reorganized the Introduction and incorporated the reference about AtSEC62 as an ERphagy receptor in plants. We have also changed in the entire manuscript the word “reticulophagy” with “ER-phagy”

2) It is important to clarify How AIM motifs within intramembrane regions of the RHD domain in Rtns proteins could be functional. How this motif is available to interact with Atg8a? Are the constructs localized in the ER membranes? Even though it was mentioned in the text and tested by co-immunoprecipitation experiments is not enough evidence to a successful insertion into the ER membranes. This is an important point to consider and clarify and also how these proteins are folded if they were mutated for alanine residues and specially in these hydrophobic regions.

To address these concerns, we have included in the revised manuscript two new assays:

1) We tested the stability of the mutant and wild type Rtn2 proteins by treating protoplasts with cycloheximide (CHX) to inhibit protein synthesis and detecting protein content by western blotting over time. These results are now included in Figure 2H. Based on this assay, we found that HARtn2(mAIM1,2,5,6) seemed to be more stable than HA-Rtn2 after 180 min of CHX treatment. However, when the same CHX chase assay was done in Arabidopsis *atg7* mutant protoplasts, which are unable to perform macroautophagy, both HA-Rtn2 and HA-Rtn2(mAIM1,2,5,6) showed similar stability. These new results indicate that a) the mutations AIM domains located at transmembrane domains do not render Rtn2 unstable; b) the identified AIM domains are required for the autophagic degradation of Rtn2 during cycloheximide treatment.

2) We performed a colocalization analysis between the three Rtn2 proteins fused to mCherry with the ER lumenal marker CFP-ER (now included in Figure 2—figure supplement 1C of the revised manuscript). We found no differences in the Pearson’s colocalization coefficients for the three Rtn2-m-Cherry fusions co-expressed with similar amounts of CFP-KDEL. Based on the fact that wild type and AIM-mutated Rtn2 proteins a) show similar or even higher stability (Figure 2H); b) show similar ER localization (Figure 2—figure supplement 1C); c) are equally able to interact with Rtn1 at the ER (Figure 2—figure supplement 1B), we assume that the AIM-mutated proteins are properly folded and inserted in the ER, indicating that the AIM domains at the transmembrane domains are indeed important for the interaction with Atg8. In the text, we explain that these AIM domains are predicted to be in the transmembrane regions but in close proximity to the ER cytoplasmic face. We speculate that in highly curved ER domains, the stretching of the outer leaflet of the ER membrane could facilitate the exposure of these AIM domains to Atg8.

3) The authors mainly tested DTT as an unfolding protein response trigger. Generally, in this kind of studies, will be important to include other stressors as thapsigargin due to the important role of ER during calcium homeostasis that will be nicely correlated with the increased interaction with the chaperones under stress. These chaperones are mainly calcium sensitive. For other hand, could be important and interesting to include the stress abiotic as a model for ER stress induction. Cereal crops are usually exposed to abiotic stress that cannot be avoided. Hence interesting mechanisms of tolerance and adaptability could be evidenced when ER-phagy will be involved. Additionally, the identification of the Rtn proteins was in cereal cells, could be possible that the function of these proteins change between the different tissues? i.e. mesophyll cells protoplasts versus aleurone or endosperm cells.

In the plants, chemical-induced ER stress is usually triggered by TM or DTT. We have used both drugs in our manuscript and found similar results (Figure 2—figure supplement 2D and E). Thapsigargin is a specific inhibitor of most animal SERCA (sarcoplasmic/endoplasmic reticulum Ca^2+^ ATPase-type) Ca^2+^pumps. Whereas thapsigargin has been reported to affect root gravitropism in Arabidopsis (Urbina et al. 2006, Biol Res 289-296), it is unclear what its specific molecular targets are in plants and where they localize. For example, thapsigargin has been reported to inhibit Ca^2+^-ATPase activity in Golgi vesicles isolated from pea epicotyls (Ordenes et al., 2002, Plant Physiol 129: 1820-1828) and to inhibit the AtHMA1 Calcium/heavy metal pump that localizes to the chloroplast envelope (Moreno et al. 2008, J. Biol Chem 283:9633). Based on the limited current knowledge of thapsigargin action in plant cells, we consider that in our study, it is best to trigger ER stress with TM or DTT. We agree with the reviewer that studying the function of Rtn1 and Rtn2 in multiple tissues and different abiotic stress conditions is very relevant and we plan to follow these lines of research in future studies.

4) The authors are advised to follow the nomenclature of published receptors that mediate ER remodelling. The process of reticulophagy was described some 10 years ago as a response of ER to the stress. The work of Walter's lab coined the nomenclature ER-phagy by describing the details of genetic and molecular characterization of this process. Since 2015 there are 8 distinct receptors that can mediate ER-phagy published including those from the yeast, plants and mammalian systems. In each manuscript the nomenclature was that these receptors were called ER-phagy receptors. Thus stating now that RTN1 and 3 from maize are reticulophagy receptors only contributes to the confusion of the readers who are not directly from this field.

We have changed the term “reticulophay” with “ER-phagy” in our manuscript, as suggested by the reviewer.

General technical comments1) To improve the manuscript the authors should reorganize the Introduction and the Figure 1 including some of the recent work published and the phylogeny analyses.

We have reorganized the Introduction and included more recent references.

2) As was mentioned before will be important to include some tools to measure the autophagy flux and specific could use the same mCherry-GFP Rtn proteins as a reporter for ER-phagy.

We have estimated autophagic flux in maize aleurone cell in two ways: 1) Detecting GFP/YFP fluorescence signal from GFP/YFP-Atg8 delivered to the vacuole by autophagy (Figure 6A-I), and 2) GFP-Atg8 cleavage assay in western blots in which we calculate the ratio free GFP to GFP-ATG8 (Figure 6J-K).

3) There are some figures with low quality and with visible resolution problems in its pictures. Also, should be more clear if the authors can prepare some magnifications according to show nicely the morphology of vacuoles and the puncta formation positive to Atg8a.

This was an unintended consequence of embedding the figures in the text for the original submission. We are now submitting high-resolution versions of all figures.

4) Subsection “Rtn1 and Rtn2 regulate macroautophagy” it is not clear the conclusion. The authors should clarify this point where the deletion of rtn2-2 in a background defective in atg12-2 affects the vacuole contains. However in the text is indicated that the loss of Rtn2 increases Atg8 macro-autophagy. These data are not correlated with the western blots of the YFP-Atg8a for example.

The loss of Rtn2 function induces an increase in autophagic flux as shown by the increased ratio free YFP to YFP-Atg8 in the *rtn2* mutant (Figure 6J). This is consistent with the increased accumulation of autophagic cargo inside *rtn2* mutant vacuoles (Figure 5D-G). This abnormal vacuolar accumulation in the *rtn2* mutant is dependent on canonical macroautophagy since when the *rtn2* mutation is in the *atg12* background (unable to perform Atg8-dependent macroautophagy), we do not observe abnormal vacuolar contents or delivery of YFP-Atg8 to the vacuoles.

[Editors' note: further revisions were suggested prior to acceptance, as described below.]

The manuscript has been significantly improved, and we are moving toward acceptance, but there are some remaining issues that need to be addressed in the text before acceptance, as outlined below:1) One statement that is still confusing and not clearly answered is how can it be that the interaction between ATG8 and RTN1/2 is mediated via the AIM that is localized into the transmembrane region of RTNs? Reviewers felt this was still not addressed in the revision.

We have now modified the Discussion to explain more clearly our hypotheses on how the AIMs at transmembrane domains of Rtn1/2 interact with Atg8.

“We identified four functional AIMs in maize Rtn1/2, one located within the C-terminal tail similar to FAM134B, one in the cytoplasmic loop, and two additional AIMs in transmembrane regions. The presence of two functional AIMs at the transmembrane segments of Rtn1/2 is puzzling. However, these AIMs are predicted to be near the cytoplasmic face of the ER membrane and therefore, could be partially accessible to Atg8 for binding. In addition, reticulon proteins are known to induce membrane stretching, which could increase the accessibility of transmembrane domains to cytosolic proteins. A recent study of FAM134B shows that the two wedge-shaped transmembrane helical hairpins of the RHD domain tend to create membrane asymmetry by stretching the cytoplasmic leaflet of the ER (Bhaskara et al., 2019). This membrane deformation is enhanced by the oligomerization of FAM134B mediated by interactions through the transmembrane domains. One could envision that membrane stretching on the cytoplasmic leaflet of the ER membrane mediated by the RHD domain induces a change in the packing of phospholipids around the transmembrane domains, increasing access of Atg8 to the transmembrane AIMs in Rtn1/2 as the ER membranes become increasingly curved (Figure 7C).”

We also add a sentence closing the first paragraph of the Discussion to highlight the potential relevance of the multiple AIM domains in Rtn1/2.

“Why Rtn1/2 harbor multiple AIMs is unclear but their presence could help these reticulons maintain contact with Atg8-decorated autophagic membranes as the ER pinches off in small fragments during ER-phagy.”

2) It was suggested that in Figure 1, because Maize Rtn1 and Rtn2 are reticulon proteins. The diagrams comparing the position of the AIM domains at least with the mammalian RTN3L, yeast Atg40 and Maize Rtn1/2 could be added in the main figure.

We have now moved the diagram comparing RTN3L, Atg40 and Rtn1/2 from Figure 1 supplement 2 to Figure 1 (panel A). The text, including figure legends, was modified accordingly.

3) Reviewers felt that it was not truly demonstrated that an increase in the Atg8 protein levels is a consequence of increased autophagy directly rather than that ER-stress triggers the upregulation of Atg8 and autophagy core machinery genes (Figure 5-6). This caveat should be made.

There seems to be some confusion regarding the analysis of Atg8 in Figures 5 and 6.

Figure 5K shows a western blot of endogenous Atg8 and other proteins (autophagy receptor Nbr1, proteasome subunit Rpt4, ER protein Aca2, mitochondrial protein Vdac, histone H3) from endosperm samples of different maize genotypes (W22, three independent *rtn2* mutants, *atg12*, and *atg12 rtn2* double mutant). Atg8 (and Nbr1) only accumulated in plants with a mutation of the core autophagy component *Atg12*. The impaired degradation of Atg8 and other autophagic components (e.g. Nbr1) has been previously shown in *atg12* and other core *atg* mutants in maize and Arabidopsis (McLoughlin et al., 2019, Jung et al., 2020) and is due to the inability to deliver Atg8-decorated autophagosomes to the vacuole. The *rtn2* mutant aleurone samples, although exhibiting ER stress (Figure 7B), did not show abnormal accumulation of Atg8. The manuscript says:

“Consistent with most aspects of autophagy being normal in the rtn2 mutant lines, we failed to see increases in the abundance of Atg8 and the autophagy receptor Nbr1 in either aleurone or starchy endosperm cells from the rtn2-1, rtn2-2, and rtn2-3 plants, while both proteins hyper-accumulated in endosperm samples from the atg12-2 and rtn2-2 atg12-2 plants (Figure 5K).”

Therefore, we do not claim that the increase in Atg8 abundance in the maize *atg12* mutant is a consequence of increased autophagy but to impaired autophagy as demonstrated previously by our groups (Li et al., 2015); the *rtn2* mutants do not show abnormal accumulation of endogenous Atg8 compared to controls.

Figure 6J shows a western blots of cleaved YFP and full length YFP-Atg8 expressed in W22 (control), *rtn2, atg12* and *rtn2 atg12* endosperm samples. Figure 6K is the quantification of the YFP to YFP-Atg8 band intensities based on three western blots. Here we see increased relative abundance of cleaved YFP (derived from YFP-Atg8 delivered to vacuoles) in relationship to full length YFP-Atg8 (in the cytoplasm) in the *rtn2* mutant aleurone cells, which means that a larger proportion of the YFP-Atg8 pool is sent to the vacuole via autophagy in *rtn2* than in W22 aleurone cells. The cleaved YFP fragment is only seen in cells that are able to perform autophagy and, as expected, is absent in *atg12* and the *atg12 rtn2* mutant samples (Figure 6K).

This ratiometric assay is widely used in animal cells, plants and yeast and it was developed in the first place to be able to quantify autophagic flux independently of the amount of Atg8 in tissues/cell. In addition, the YFP-Atg8 reporter is not driven by its endogenous promoter but by the maize UB10 promoter. We have not observed changes in the abundance of this reporter in the different genotypes.